# The Development and Assessment of a Unique Disulfidptosis-Associated lncRNA Profile for Immune Microenvironment Prediction and Personalized Therapy in Gastric Adenocarcinoma

**DOI:** 10.3390/biomedicines13051224

**Published:** 2025-05-19

**Authors:** Jiyue Zhu, Xiang Zhu, Tingting Su, Huiqing Zhou, Shouhua Wang, Weibin Shi

**Affiliations:** 1Department of General Surgery, Xinhua Hospital, Shanghai Jiao Tong University School of Medicine, Shanghai 200092, China; zhujiyue2022@163.com (J.Z.); zhusurg@163.com (X.Z.); sistine_9191@163.com (T.S.); 2Shanghai Key Laboratory of Biliary Tract Disease Research, Shanghai 200092, China; 3Department of Gastroenterology, Xinhua Hospital, Shanghai Jiao Tong University School of Medicine, Shanghai 200092, China; hq.zhou@yahoo.com

**Keywords:** gastric cancer, disulfidptosis, lncRNA, TCGA, prognostic model, immune status

## Abstract

**Background:** Long non-coding RNAs (lncRNAs) are crucial factors affecting the occurrence, progression, and prognosis of gastric carcinoma (GC). The accumulation of disulfide bonds to excessive levels in cells expressing high *SLC7A11* triggers disulfidptosis, which functions as a regulated form of cellular death. Research has demonstrated that upregulated *SLC7A11* is common in human cancers, but the effect of disulfidptosis on GC remains unclear. Identifying lncRNAs associated with disulfidptosis (drlncRNAs) and establishing a prognostic risk profile holds considerable importance for advancing GC research and treatment. **Methods:** Clinical records and transcriptomic datasets from individuals with GC were acquired from The Cancer Genome Atlas (TCGA) repository. A three-drlncRNA risk model was built using three common regression analysis methods. Then we used receiver operating characteristic (ROC) curves, independent prognostic analysis, and additional statistical approaches to assess the precision of the model. This investigation additionally encompassed Gene Ontology (GO) and Kyoto Encyclopedia of Genes and Genomes (KEGG) analysis, immune cell infiltration evaluation, and pharmacological sensitivity predictions. To further investigate immunotherapy response disparities between patient cohorts with elevated- and reduced-risk scores, analyses of tumor mutational burden (TMB), tumor immune dysfunction and exclusion (TIDE), and microsatellite instability (MSI) were implemented. **Results:** We constructed a unique model composed of three drlncRNAs (AC107021.2, AC016394.2, and AC129507.1). Its independent prognostic capability for GC patients was validated through both single-variable and multivariable Cox regression analyses. GO and KEGG pathway assessments revealed predominant enrichment within the elevated-risk cohort, particularly in pathways involving sulfur compound interactions, traditional Wnt signaling mechanisms, cell-substrate adherens junctions, and cAMP signaling cascades, among others. Tumor microenvironment (TME) evaluation demonstrated elevated ImmuneScores, StromalScores, and ESTIMATEScores within the high-risk patient population. Concurrently, this elevated-risk cohort exhibited enhanced immune cell infiltration patterns, whereas the reduced-risk group displayed superior expression of immune checkpoints (ICPs). Additional investigations revealed that patients categorized into the reduced-risk classification possessed greater tumor mutational burden, increased MSI-high proportions, and diminished tumor immune dysfunction and exclusion scores compared to their high-risk counterparts. Pharmacological sensitivity assessments confirmed the superior efficacy of several therapeutic agents, including gemcitabine and veliparib (ABT.888), in patients with lower risk classifications. **Conclusions:** Our established risk stratification system demonstrates independent prognostic predictive capacity while offering personalized clinical intervention guidance for individuals diagnosed with GC.

## 1. Introduction

Globally, gastric carcinoma (GC) ranks among the most prevalent malignancies, with annual worldwide incidence approaching 1 million new diagnoses and mortality exceeding 720,000 individuals annually [1]. Epidemiological data from 2020 documented 1.1 million newly identified cases and approximately 770,000 fatalities attributed to this gastrointestinal malignancy worldwide. Projections suggest a substantial escalation in these figures, with epidemiological forecasts anticipating approximately 1.8 million additional diagnoses and 1.3 million GC-related mortalities by 2040 [2]. Early GC typically presents with subtle and nonspecific symptoms, resulting in more than 60% of patients being diagnosed after metastasis [3]. This late diagnosis contributes to poor prognosis, with only 5% of patients with metastatic GC surviving beyond five years [4,5].

Fortunately, therapeutic advances in immunological interventions, particularly immune checkpoint blockade (ICB) agents, have offered promising approaches for managing progressive and relapsed gastric malignancies [6]. Nevertheless, a considerable percentage (40–60%) of individuals receiving immune checkpoint (ICP)-targeted therapies exhibit therapeutic resistance [7]. Consequently, both the identification of dependable prognostic indicators to anticipate survival outcomes in gastric adenocarcinoma (GA) patients and the recognition of treatment-responsive populations and effective pharmaceutical agents hold paramount importance for ultimately enhancing longevity among individuals with GC.

Disulfidptosis is a kind of original regulated cell death that is different from ferroptosis and cuproptosis. The initial characterization by Liu and colleagues established disulfidptosis as a distinct cellular death mechanism triggered by disulfide bond overaccumulation within cells expressing elevated levels of *SLC7A11* [8]. Here we focus on long non-coding RNAs associated with disulfidptosis (drlncRNAs) related to 24 disulfidptosis-related genes (DRGs) mentioned in the literature, including *GYS1*, *NDUFS1*, *OXSM*, *LRPPRC*, *NDUFA11*, *NUBPL*, *NCKAP1*, *RPN1*, *SLC3A2*, *SLC7A11*, *INF2*, *CD2AP*, *ACTN4*, *PDLIM1*, *IQGAP1*, *DSTN*, *CAPZB*, *ACTB, MYL6*, *MYH9*, *MYH10*, *TLN1*, *FLNA*, and *FLNB*. Among these 24 DRGs, the last 14 are genes that encode actin-related proteins and are upregulated by glucose starvation [8]. Given that upregulated *SLC7A11* is common in human cancers, the discovery of disulfidptosis expands the framework of programmed cell death and may lead to new treatment and therapeutic targets for multiple cancers [9,10,11].

Expanding the scientific literature demonstrates that lncRNAs functioning as oncogenic elements can enhance the proliferative capacity, invasive potential, and metastatic dissemination of GC cells, while simultaneously serving as prospective molecular indicators for survival prognostication and personalized therapeutic decision-making in GA patients [12,13,14,15]. However, drlncRNAs in GA have barely been reported so far.

This investigation endeavors to establish an innovative lncRNA profile associated with disulfidptosis for prognostic assessment and immune microenvironment characterization in GA. We anticipate that our findings will contribute significant insights toward the identification of efficacious therapeutic compounds and the selection of patients most likely to benefit from immunological interventions.

## 2. Materials and Methods

### 2.1. Data Source

The methodological framework of this investigation is illustrated in Figure 1. Transcriptomic datasets comprising 448 GA cases (412 neoplastic specimens and 36 non-neoplastic tissues) along with clinical information from 443 patients were acquired from The Cancer Genome Atlas (TCGA) repository (https://tcga-data.nci.nih.gov/tcga (accessed on 21 March 2023)). For differentiation between mRNAs and lncRNAs, we utilized the Strawberry Perl computational platform (version 5.30.0-64bit) obtained from https://strawberryperl.com. A total of 24 DRGs were the same as those used in the study by Liu et al., and Pearson correlation analysis was then performed to filter out drlncRNAs [8]. In order to minimize errors, patients with missing survival information were excluded and 407 GA patients remained. Simple nucleotide variation (SNV) profiles from 434 GA subjects were additionally extracted from the TCGA repository and subsequently employed for tumor mutational burden (TMB) quantification. The data regarding GA microsatellite status were obtained from The Cancer Immunome Atlas (TCIA) website (https://www.tcia.at (accessed on 21 March 2023)). Tumor immune dysfunction and exclusion (TIDE) metrics were procured from the Harvard Dana-Farber Cancer Institute’s online platform (http://tide.dfci.harvard.edu (accessed on 21 March 2023)).

### 2.2. The Development and Verification of the drlncRNAs Signature

The comprehensive cohort encompassing 407 GA cases underwent random allocation into training and validation datasets at an equal 1:1 distribution ratio. Then, statistical analysis was implemented to substantiate the classification robustness. We first developed the drlncRNA prognostic model in the training set. Application of univariate Cox proportional hazards regression identified 29 drlncRNAs that exhibited substantial association with overall survival (OS) (*p* < 0.05). To mitigate potential overfitting issues, we subsequently implemented the least absolute shrinkage and selection operator (LASSO)-penalized Cox methodology with 10-fold cross-validation under identical significance thresholds (*p* < 0.05), yielding seven candidate transcripts. Multivariable Cox regression analysis further refined this selection, culminating in a prognostic algorithm incorporating three drlncRNAs. The model’s performance underwent verification utilizing both the validation cohort and the aggregate patient population. Risk stratification scores were derived according to the following mathematical expression:Risk Score = Σ (i = 1 to n) Coef(i) × Expr(i).

Coef(i) is the coefficient of the i-th lncRNA.

Expr(i) is the expression level of the i-th lncRNA.

Based on the median computed risk value as the threshold criterion, all GA subjects were stratified into elevated or reduced disulfidptosis score categories. The prognostic algorithm’s precision was evaluated through independent survival analyses to determine whether this drlncRNA profile could predict clinical outcomes in GC patients autonomously from demographic variables (age and gender) and tumor characteristics (histological grade and pathological stage). Subsequently, we used the R (version 4.2.1-64bit) packages “survival”, “survminer”, and “timeROC” to plot the receiver operating characteristic (ROC) curves of the training set, the test set, and the entire set as well as different clinicopathological characteristics, respectively, and the corresponding area under the curve (AUC) values were also calculated. To further corroborate our prognostic algorithm’s efficacy, survival analyses employing the Kaplan–Meier methodology were conducted across various clinicopathological parameters including patient age, histological differentiation, disease progression stage, primary tumor dimensions (T classification), regional lymphatic involvement (N classification), and remote metastatic status (M classification).

### 2.3. Independent Prognostic Analysis and ROC Curve Plotting

Independent prognostic analysis was utilized to confirm that our model can predict the outcomes of patients independent of age, gender, stage, and grade. The predictive precision of the prognostic model was assessed through ROC curve and AUC values (the R packages “limma”, “scatterplot3d”, “survival”, “survminer”, and “timeROC” were used in this process).

### 2.4. Nomogram and Calibration

Considering clinicopathological factors and risk score, a predictive nomogram that was designed to predict 1-, 3-, and 5-year OS was created utilizing the R packages “survival”, “regplot”, and “rms”. In the meantime, calibration curves for 1-, 3-, and 5-year survival projections were generated to assess the predictive accuracy of the nomogram.

### 2.5. GO and KEGG Enrichment Analysis

The Gene Ontology (GO) repository categorizes genetic functions into three distinct classifications: biological process (BP), cellular component (CC), and molecular function (MF). Utilizing the “clusterProfiler” R computational package, we performed GO and Kyoto Encyclopedia of Genes and Genomes (KEGG) pathway enrichment analyses to elucidate potentially enriched functional categories and signaling cascades, with subsequent visualization implemented through the “GOplot” and “ggplot2” R visualization libraries.

### 2.6. Investigation of Tumor Microenvironment, Immune Infiltration, and Immune Checkpoints

Based on the results of GO and KEGG, we conducted immune-related analyses. The Estimation of STromal and Immune cells in MAlignant Tumor tissues using Expression data (ESTIMATE) algorithm is usually utilized to determine the immune, stromal, and comprehensive scores of the tumor microenvironment (TME) [16]. The Estimation of the STromal and Immune cells in MAlignant Tumor tissues using Expression data (ESTIMATE) algorithm was implemented through the ‘limma’ and ‘ggpubr’ R packages to investigate potential associations between risk stratification scores and TME characteristics. The differential infiltration patterns of specific immune cellular subsets between elevated- and reduced-disulfidptosis score groups underwent examination via seven distinct analytical platforms—XCELL, TIMER, QUANTISEQ, MCPCOUNTER, EPIC, CIBERSORT, and CIBERSORT-ABS—with the results depicted through bubble plot visualization. Multiple R packages including “limma”, “scales”, “ggplot2”, “ggtext”, “reshape2”, “tidyverse”, and “ggpubr” facilitated this analytical process. Subsequently, quantitative assessment of 29 immune cell populations and functional parameters across 448 GA specimens was performed by utilizing the single-sample gene set enrichment methodology (ssGSEA) through the “GSVA”, “limma”, and “GSEABase” R packages, with the resultant data visualized via dual boxplot representations using the “limma”, “ggpubr”, and “reshape2” software packages. Finally, we conducted an analysis to explore differentially expressed ICPs between high- and low-disulfidptosis score groups.

### 2.7. TMB, MSI, and TIDE

Initially, simple nucleotide variation profiles from 434 GA patients were acquired from the TCGA repository, with the subsequent tumor mutational burden quantification executed through the Strawberry Perl computational platform. Mutational landscape visualization via waterfall diagrams depicting the fifteen most frequently altered genes in elevated- versus reduced-risk cohorts was generated using the “maftools” package. Survival analysis curves stratifying patients into four distinct categories (“elevated TMB with high risk”, “elevated TMB with low risk”, “reduced TMB with high risk”, and “reduced TMB with low risk”) were constructed by utilizing the “survival” and “survminer” R libraries. Additionally, based on extracted GA microsatellite stability data, graphical representations including bar charts and boxplots were developed to elucidate the associations between microsatellite instability (MSI) status and risk classification scores. Ultimately, a comparative analysis of tumor immune dysfunction and exclusion metrics between patient populations with differential disulfidptosis score classifications was performed.

### 2.8. Drug Sensitivity Analysis

Inhibitory concentration-50 (IC50) measurements for diverse therapeutic agents were determined and juxtaposed between elevated- and reduced-disulfidptosis score patient cohorts through implementation of the “pRRophetic” R computational package, with resultant comparative analyses displayed as boxplot representations.

## 3. Results

### 3.1. Data Source and Processing

Using Strawberry Perl software (version 5.30.0-64bit), we processed the TCGA transcriptome data (raw gene count data were generated using the STAR alignment tool, with the file name suffix ‘augmented_star_gene_counts.tsv’) and gained the expression data of 19,938 mRNAs and 16,876 lncRNAs. This dataset contained unnormalized read counts mapped to each gene, representing the original number of reads aligned to each gene region before any normalization based on gene length or sequencing depth, such as Fragments Per Kilobase of transcript per Million mapped reads (FPKM) or Transcripts Per Million (TPM). By combining the mRNA expression profiles with 24 DRGs, we obtained the expression data of 24 DRGs. Then, based on the criteria of |Pearson R| > 0.4 and *p* < 0.001, a Pearson correlation analysis of 16,876 lncRNAs and 24 DRGs was conducted and 367 drlncRNAs were finally identified (Figure 2A).

### 3.2. Development and Verification of drlncRNA Predictive Algorithm

By combining the expression data of 367 drlncRNAs with the clinical information of 407 GC patients, we acquired the drlncRNA expression data profile and survival data for 407 samples, which were used for risk model building. We first implemented univariate Cox regression analysis and preliminarily identified 29 drlncRNAs (Appendix A, *p* < 0.05). Subsequently, LASSO-penalized Cox analysis was implemented to mitigate model overfitting concerns and 7 drlncRNAs significantly related to OS were obtained (Figure 2B, *p* < 0.05). Eventually, the drlncRNA risk signature, comprising three lncRNAs (*AC107021.2*, *AC016394.2*, and *AC129507.1*) and the corresponding risk coefficient, was acquired (Figure 2C, *p* < 0.05). In the meantime, the relationship between 24 DRGs and 3 drlncRNAs is presented in Figure 2D. According to the previous calculation formula, Risk Score=0.660×AC107021.2_exp−0.436×AC016394.2_exp+0.637×AC129507.1_exp. In this formula, *AC107021.2*_exp, *AC016394.2*_exp, and *AC129507.1*_exp represent the expression levels of lncRNAs *AC107021.2*, *AC016394.2*, and *AC129507.1*, respectively.

A total of 407 GA samples were then allocated into two groups randomly at a 1:1 ratio. The dataset distribution consisted of 204 specimens allocated to the training cohort and 203 cases assigned to the validation group. Based on the median risk score as a cutoff value, patients were divided into high- and low-risk groups. According to the risk score and survival status of the training set patients, manifested in Figure 3A,B, we can conclude that the mortality of the patients was positively correlated with the risk score. The Kaplan–Meier curve of the training set (Figure 3C, *p* < 0.001) indicated that the survival time of the patients in the high-risk group was notably lower than that in the low-risk group, which was consistent with our expectations. To demonstrate the predictability of the drlncRNA model, we conducted the same analysis in the test set and the whole set. Findings derived from both the validation cohort (Figure 3D–F) and the aggregate patient population (Figure 3G–I) demonstrated concordance with the results observed in the TCGA training dataset.

Constructed by the model, the risk heatmaps of the three drlncRNAs (Figure 4A–C) show that the upregulated *AC107021.2* and *AC129507.1* lncRNAs were significantly associated with high-disulfidptosis score patients, while *AC016394.2* was upregulated in the low-disulfidptosis score patients. Independent prognostic evaluation demonstrated that the three drlncRNA signatures constitute a robust survival prediction tool, functioning autonomously from demographic and clinical parameters including patient age and disease stage (Figure 4D,E).

Subsequently, ROC curve analysis yielded AUC measurements within the training population (1-year AUC = 0.741, 3-year AUC = 0.654, and 5-year AUC = 0.781; Figure 5A), validation population (1-year AUC = 0.638, 3-year AUC = 0.640, and 5-year AUC = 0.707; Figure 5B), and comprehensive cohort (1-year AUC = 0.692, 3-year AUC = 0.646, and 5-year AUC = 0.723; Figure 5C). AUC values for ROC analyses across various clinicopathological features were also calculated, and the ROC curve predicated upon our risk model showed the greatest AUC value (Figure 5D), which illustrates that our risk model is considerably accurate as a prognosis predictor.

To further substantiate the predictive reliability of our algorithm, Kaplan–Meier survival analyses were generated that compared patient outcomes between elevated- and reduced-risk classifications across various clinicopathological parameters including patient age, histological differentiation grade, disease progression stage, primary tumor extent (T classification), nodal involvement status (N classification), and metastatic disease presence (M classification). We can recognize that our prognostic signature can accurately prognosticate the survival of patients, independent of clinical variables (Appendix A). For patients in the M1 stage, there was no significant difference in survival between our high- and low-risk groups, which may be due to the lack of a sufficient number of samples.

### 3.3. Nomogram and Calibration Curves

To facilitate survival duration estimation, utilizing our drlncRNAs profile, we constructed a nomographic prediction model capable of projecting 1-, 3-, and 5-year survival probabilities (Figure 6A). Concurrently, calibration plot analyses were developed to validate the predictive precision of this nomographic representation (Figure 6B).

### 3.4. Functional Analysis of Model

The circle chart and corresponding notes (Figure 7A,B), column chart (Figure 7C), and bubble chart (Figure 7D) of GO and KEGG enrichment analysis revealed significant associations between the three drlncRNAs (*AC107021.2*, *AC016394.2*, and *AC129507.1*) and the sulfur compound-binding, canonical Wnt signaling pathway, the cell-substrate adherens junction, and the cAMP signaling pathway. In the process of disulfidptosis, sulfur compounds play a crucial role. These sulfur-containing molecules primarily include cysteine, cystine, glutathione (GSH), thioredoxin (TXN), and their derivatives (such as disulfides), which participate in cellular redox regulation through sulfur atoms [17,18]. Specifically, *SLC7A11*-mediated cystine uptake is the initiating step of disulfidptosis, while GSH, as a major antioxidant, leads to disulfide stress when depleted. Simultaneously, dysregulation of the TXN system directly induces disulfidptosis. At the molecular level, “sulfur compound binding” primarily manifests as proteins interacting with sulfur-containing molecules through disulfide cross-linking (such as *SLC7A11* binding to cystine [19,20]), enzyme-substrate binding (such as thioredoxin reductase *TXNRD1* binding to oxidized TXN [21,22]), and antioxidant regulation (such as covalent binding of glutathione transferase GST to GSH [23,24]). Ultimately, this process triggers cytoskeletal collapse and cell death through excessive cystine uptake caused by *SLC7A11* overexpression, sulfur metabolism imbalance induced by GSH depletion, and disulfide cross-linking of key proteins such as actin (ACTB) [8,25,26].

### 3.5. Immune Infiltration Status

For the purpose of exploring the difference in the TME between two disulfidptosis score groups, we implemented the ESTIMATE computational approach to quantify neoplastic tissue purity. According to the boxplots, quantitative assessment revealed that elevated-risk cohorts exhibited superior scoring metrics compared to their reduced-risk counterparts in terms of StromalScore (*p* = 2.5 × 10^−10^), ImmuneScore (*p* = 0.039), and ESTIMATEScore (*p* = 6.1 × 10^−6^) (Figure 8A). Subsequently, we analyzed the differences in the immune cell subpopulations in TME between the two risk groups. Bubble plot visualization demonstrated that patient cohorts with elevated risk scores corresponded to enhanced immune cell infiltration density, such as the presence of cancer-associated fibroblast (CAF) in the XCELL [27], the MCPCOUNTER [28], and the EPIC [29], and the presence of macrophage M2 in the QUANTISEQ [30], the CIBERSORT, and the CIBERSORT-ABS [31] (Figure 8B). Additionally, boxplot analyses comparing 29 immune cellular populations and functional parameters from risk-stratified cohorts indicated that patients with an elevated risk classification exhibited increased mast cell infiltration coupled with diminished major histocompatibility complex class I (MHC-I) expression levels (Figure 8C,D). ICP molecule evaluation demonstrated upregulated expression patterns of several checkpoints including programmed cell death protein 1 (PDCD-1), CD274 (programmed cell death ligand 1) (PD-L1), and cytotoxic T lymphocyte-associated antigen 4 (CTLA-4) within the reduced-risk population (Figure 8E).

### 3.6. Immunotherapy Response Analysis

At this stage, TMB, MSI, and TIDE analyses were conducted. TMB comparative analysis demonstrated elevated genomic alteration frequencies among patients in the reduced-risk category (Figure 9A). To comprehensively investigate genetic mutational landscape disparities between risk-stratified cohorts, we generated corresponding mutational waterfall visualizations. Within the fifteen most frequently mutated genes, only the *TP53* tumor suppressor gene exhibited increased mutational prevalence in the high-disulfidptosis score patients, and we noticed that the mutation rate of the ARID1A gene in the high-risk cohort exhibited substantially diminished prevalence compared to patients in the reduced-risk population (Figure 9B). Next, we explored the relationship between TMB, patient survival, and risk score (Figure 9C). Survival analysis revealed superior outcomes among GC patients exhibiting an elevated tumor mutational burden compared to those with reduced genomic alteration frequencies (*p* = 0.009), with the mutational burden status demonstrating independent prognostic significance irrespective of risk stratification metrics. In addition, when we explored the relationship between MSI and risk score, we found that the proportion of patients with “MSI-H (microsatellite instability-high)” in the low-disulfidptosis score group (32%) was much higher than that in the high-disulfidptosis score group (9%) (Figure 9D). The risk score of patients with “MSI-H” was much lower than that of patients with “MSS (microsatellite stability)” (*p* = 4.7 × 10^−10^) and “MSI-L (microsatellite instability-low)” (*p* = 1.7 × 10^−5^) (Figure 9E). Finally, TIDE computational analysis demonstrated elevated immune evasion scores among the high-disulfidptosis cohort (Figure 9F).

### 3.7. Drug Sensitivity Analysis

In our drug sensitivity analysis, we screened a total of 42 drugs that showed differential sensitivity between high- and low-risk groups. Among these, 17 drugs demonstrated higher sensitivity in the low-risk population, while the other 25 drugs were more effective in the high-risk group. In Figure 10, we chose to display 16 drugs (e.g., gemcitabine, ABT.888) that showed enhanced sensitivity in the low-risk population to highlight potential therapeutic options that might be more effective for low-risk patients.

## 4. Discussion

A literature search revealed 24 genes involved in the disulfidptosis pathway. Co-expression analysis enabled us to identify lncRNAs associated with disulfidptosis. Subsequently, using the Cox proportional hazards regression methodology combined with LASSO penalty analysis, we constructed a prognostic signature comprising three drlncRNAs. *AC107021.2* has emerged as a diagnostic marker for gastric and lung adenocarcinomas [32,33], while *AC016394.2* and *AC129507.1* show diagnostic potential in both GA and prostate carcinoma [34,35]. Nevertheless, the interrelationship between drlncRNA molecules and immunological microenvironment characteristics in GA, along with their prognostic significance, remains to be elucidated. Our study investigated the role of drlncRNAs in GC and established a three-drlncRNA prognostic signature to predict the prognosis and provide precise and individual clinical treatment guidance for patients with GC.

Through independent prognostic analysis, we demonstrated that the ability of our risk model to predict patient outcomes remains significant when accounting for other clinical parameters. ROC curve analyses yielded AUC measurements, thereby confirming our risk stratification algorithm’s prognostic precision for patient mortality prediction across 1-, 3-, and 5-year timeframes. Kaplan–Meier curves of different clinicopathological parameters including patient age, histological differentiation, disease progression stage, primary tumor dimensions, regional lymphatic involvement, and remote metastatic status illustrated that our risk model can predict patient outcomes regardless of clinical variables. Through nomogram and calibration curves, our model enables individualized prediction of patient survival at 1-, 3-, and 5-year timepoints.

As anticipated, enrichment analysis revealed a strong association between drlncRNAs and sulfur compound binding. The identification of disulfidptosis revealed a promising therapeutic strategy for cancer treatment, wherein sulfur compound binding orchestrates this cell death process through the activation of specific signaling cascades [10]. GO and KEGG analysis also revealed other functions and pathways that are closely related to drlncRNA, such as G protein-coupled receptor (GPCR) binding and the cAMP signaling pathway. A recent study revealed that activation of the Gαs-PKA (GPCR–Gαs–PKA) signaling pathway drives CD8⁺ T cell dysfunction and confers resistance to immunotherapy [36]. cAMP exhibits dual regulatory effects on tumor cell survival and proliferation through its functional interplay with diverse immunological constituents within TME, particularly T cells and tumor-associated macrophages (TAMs) [37,38,39,40,41,42].

Given that GO and KEGG enrichment analyses revealed associations between drlncRNAs and immune function, we proceeded to investigate the immune landscape by comparing differential patterns of immunological cellular infiltration between patient cohorts stratified by contrasting risk classifications. The elevated TME scores observed in high-risk patients indicate enhanced immune cell infiltration and enriched stromal cell content in their TME. Macrophages exhibit two distinct polarization states, M1 and M2 [43]. In our analysis of immune cell infiltration patterns within the TME, we observed contrasting infiltration levels between M1 and M2 macrophages across the two disulfidptosis score groups—high levels of M1 macrophage infiltration were observed in the low-risk group, while high levels of M2 macrophage infiltration were observed in the high-risk group. While M1 macrophages significantly contribute to anti-tumor immunity, M2 macrophages facilitate tumor progression by promoting immune escape, angiogenesis, and extracellular matrix remodeling in tumor cells [44,45]. This observation potentially substantiates the poor survival of individuals in the high-disulfidptosis score group. In addition, the distinctions between sixteen immune cells and thirteen immune-related pathways across the two disulfidptosis score groups were investigated using ssGSEA. In the high-disulfidptosis score group, we observed elevated levels of mast cells accompanied by downregulation of MHC-I molecules. A recent study shows that mast cells contribute to resistance against anti-programmed cell death protein 1 (PD-1) immunotherapy, and targeted depletion of mast cells enhances the therapeutic efficacy of ICB [46]. Additional research shows that cancer cells can escape immune detection by reducing the expression of MHC-I. This decrease in MHC-I expression represents a key pathway for both inherent and adaptive resistance to immunotherapeutic interventions in cancer patients [47]. Unlike traditional therapy, this approach does not directly target tumor cells. Instead, it works by alleviating immunosuppression and stimulating the body’s natural anti-tumor immune response, demonstrating remarkable efficacy across various treatment-resistant tumors. When combined with chemotherapy, targeted therapy, radiotherapy, and other treatment modalities, it enhances overall therapeutic outcomes [48,49,50,51]. The elevated expressional profiles of ICP molecules, including programmed cell death protein 1 (PDCD1), PD-L1, and CTLA-4, in the low disulfidptosis group suggests a potentially enhanced clinical efficacy following immunotherapy.

TMB quantifies the cumulative frequency of nonsynonymous somatic alterations detectable throughout the neoplastic genomic landscape. As a molecular prognostic indicator for immune checkpoint inhibitor (ICI) therapy response, MSI-H demonstrates predictive value across diverse tumor types, potentially due to its role as a mechanistic mediator of immunotherapy outcomes [52]. Accumulating evidence suggests that TMB levels serve as a predictor of patient responsiveness to ICIs, particularly when combined with PD-1 expression and MSI status for enhanced prognostic accuracy [53,54,55,56]. Our analysis demonstrated that patients with low disulfidptosis scores exhibited significantly higher TMB (*p* = 1.2 × 10^−9^) and a greater proportion of MSI-H cases (32%) compared to the high-score group. Collective observations indicate that patients with low disulfidptosis scores may achieve enhanced therapeutic responses to immunotherapy. Moreover, we found that patients with elevated TMB showed improved survival outcomes regardless of disulfidptosis score. To systematically characterize the genomic alteration pattern disparities between contrasting risk stratification cohorts, we constructed mutational waterfall visualizations representing the fifteen genes exhibiting the highest modification frequencies. Notably, among the fifteen most frequently mutated genes, only the tumor suppressor gene *TP53* showed a higher mutation frequency in the high-risk group compared to the low-risk group. In contrast, we observed that *ARID1A* mutations were markedly less prevalent in patients with high disulfidptosis scores, establishing a distinct mutational pattern between the two risk classifications. *TP53* represents the most frequently mutated gene across human cancers [57]. Its mutations not only compromise its tumor-suppressive functions, but also confer oncogenic properties to the mutant *p53* protein [58]. Studies have demonstrated that *TP53* mutations correlate with poor clinical outcomes in cancer patients [59]. This observation, on the other hand, provides a mechanistic explanation for the poor survival outcomes observed in patients with high disulfidptosis scores. Multiple clinical trials have demonstrated that *ARID1A*-mutated solid tumors exhibit enhanced responsiveness to ICI interventions across diverse malignancy classifications, independent of MSI status or TMB [60,61,62]. Collective observations indicate that individuals with high disulfidptosis scores may exhibit resistance to ICI therapy.

TIDE analysis further showed that patients with high disulfidptosis scores exhibited an increased probability of developing resistance to immunotherapy, suggesting the limited therapeutic benefit of ICB in this subgroup. Moreover, comprehensive drug sensitivity analysis identified multiple potential therapeutic agents that showed enhanced efficacy in the reduced-risk classification while also revealing drugs prone to resistance in the increased-risk classification, thereby providing valuable insights into personalized treatment strategies. We specifically highlighted gemcitabine and ABT.888 (veliparib) based on their superior clinical translation potential (gemcitabine is widely used in standard treatment regimens for multiple cancers, while ABT.888 is a clinically advanced PARP inhibitor) [63,64,65], unique synergistic mechanisms with immunotherapy (gemcitabine induces immunogenic cell death and enhances antigen presentation [66,67], ABT.888 enhances immune recognition by increasing tumor mutational burden [68]), and solid clinical evidence base (both have clinical trial data regarding their use in combination with immune checkpoint inhibitors) [69,70]. While other drugs such as AZD6244, Metformin, and Epothilone B also show potential value in specific contexts, we believe that gemcitabine and ABT.888 represent the options most likely to rapidly translate into clinical practice, particularly for our study’s population of low-risk patients who might benefit from immunotherapy. Contemporary clinical investigations have established that the combination of gemcitabine plus cisplatin with PD-L1 inhibitors exhibits promising therapeutic efficacy in patients with advanced biliary tract cancer [69]. Clinical studies have confirmed that the combination of ABT.888 (veliparib) with PD-1 inhibitor plus platinum-based doublet chemotherapy demonstrates promising therapeutic efficacy in patients with metastatic or advanced non-small cell lung cancer [70]. According to these results, the combination of ICIs with conventional chemotherapeutic agents represents a promising therapeutic strategy for GC. Finally, we constructed a summary figure (Figure 11) of the research findings based on the above discussion results.

Additionally, we realized that this study still has limitations. Firstly, the three-drlncRNA prognostic model was developed and verified in light of the findings from retrospective analyses of TCGA. It is necessary to conduct prospective cohort studies to further validate this risk model. Secondly, more independent immunotherapy cohorts are needed to confirm the predictive value of the prognostic signature for immunotherapy response. Lastly, the validation of this predictive framework’s reliability and clinical implementation value necessitates the acquisition of extensive patient datasets, complemented by mechanistic investigations into how these lncRNA molecules influence GC initiation and progression. 

## 5. Conclusions

In summation, our established drlncRNA prognostic model enables accurate prediction of patient survival and facilitates the identification of potentially effective drugs and treatment-sensitive individuals, ultimately contributing to improved survival outcomes in GA patients.

## Figures and Tables

**Figure 1 biomedicines-13-01224-f001:**
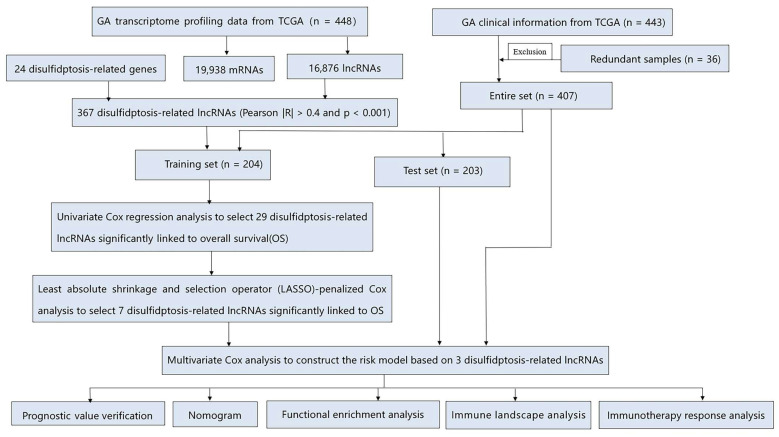
The methodological framework of the present study.

**Figure 2 biomedicines-13-01224-f002:**
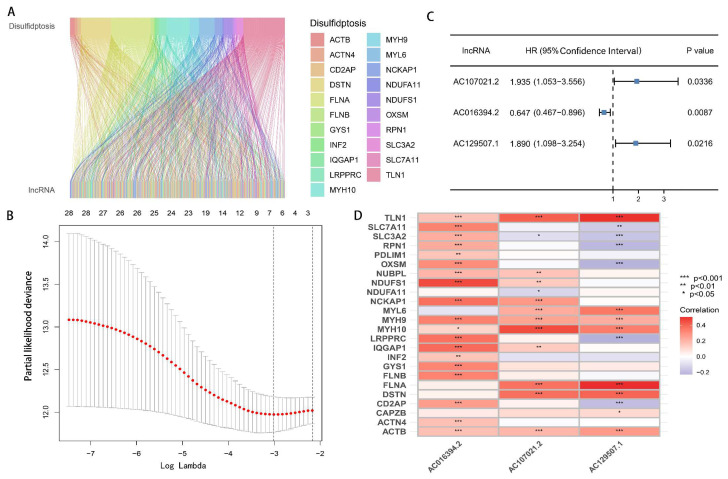
The characterization of drlncRNAs in GC and the establishment of a risk signature. (**A**) The identification of 367 drlncRNA transcripts within GA specimens. (**B**) The application of LASSO regression to the 30 overall survival-associated transcripts determined through the univariate Cox regression methodology. The top axis shows the number of non-zero features at each lambda value. Left dashed line: lambda.min (minimum cross-validation error); right dashed line: lambda.1se (one standard deviation rule). Lambda (λ) is the key regularization parameter in LASSO regression, with its optimal value determined through 10-fold cross-validation (corresponding to the dashed line in Figure 2B), which balances model complexity and variable selection. (**C**) The prognostic signature predicated upon disulfidptosis-related elements was established through multivariable Cox regression analytic approaches. (**D**) The associative patterns between the prognostic model and the DRGs.

**Figure 3 biomedicines-13-01224-f003:**
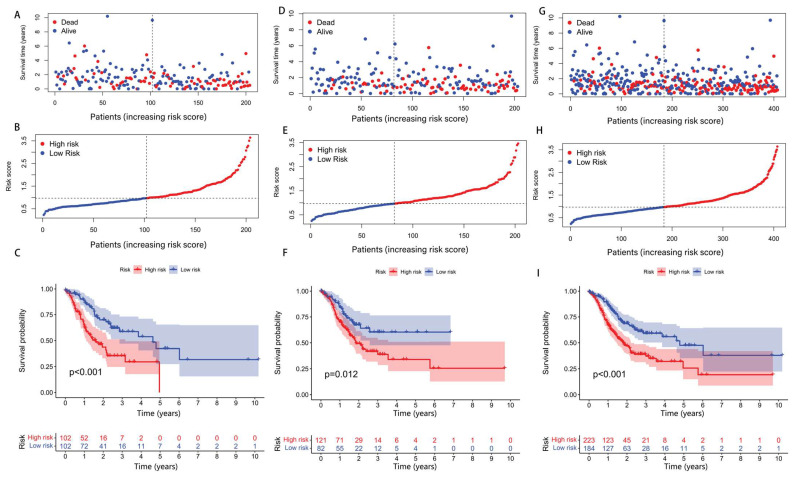
A prognostic assessment of the risk stratification algorithm across the training, validation, and comprehensive cohorts. (**A**–**C**) A comparative analysis of risk distribution, survival outcomes, and prognostic differences between the elevated- and reduced-disulfidptosis score categories within the training dataset. (**D**–**I**) Parallel analytical evaluations conducted in the validation cohort and aggregate patient population.

**Figure 4 biomedicines-13-01224-f004:**
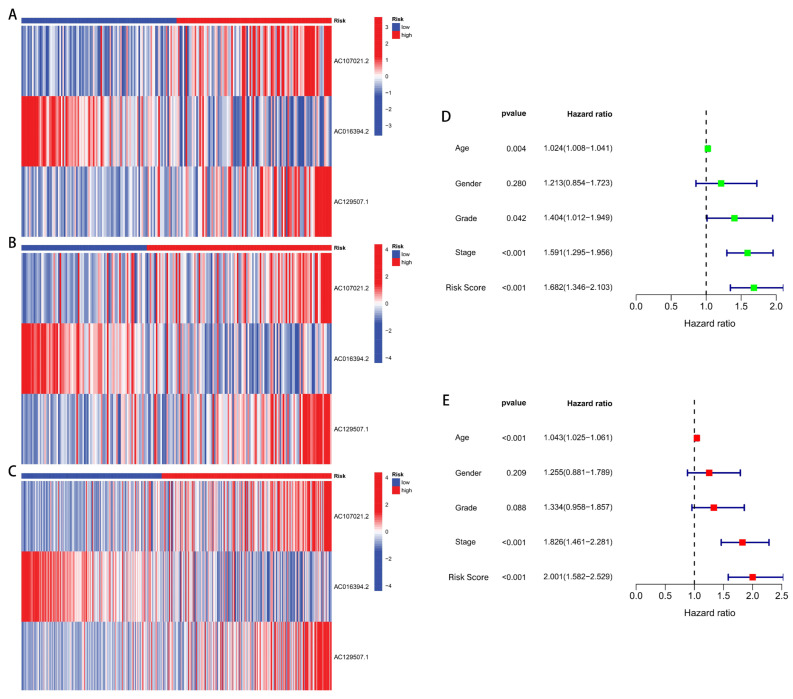
Risk heatmaps and independent prognostic analysis. (**A**–**C**) Risk heatmaps of the training, test, and entire sets. (**D**) A forest plot visualization depicting univariate Cox proportional hazards regression of diverse clinicopathological variables. (**E**) The multivariable Cox regression modeling results.

**Figure 5 biomedicines-13-01224-f005:**
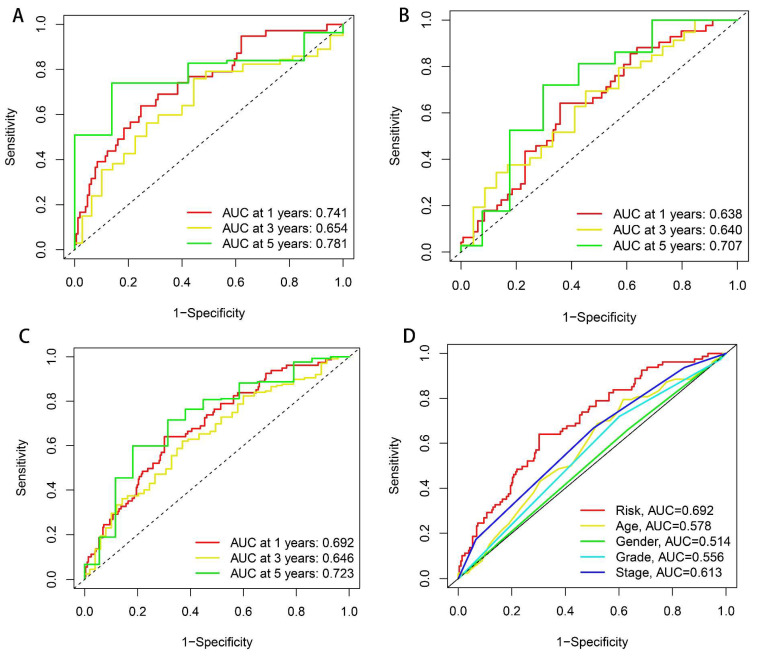
ROC curve analysis. (**A**–**C**) ROC curves representing 1-year, 3-year, and 5-year predictive performance in the training dataset, validation cohort, and comprehensive patient population. (**D**) ROC curves comparing drlncRNA models with different clinical features.

**Figure 6 biomedicines-13-01224-f006:**
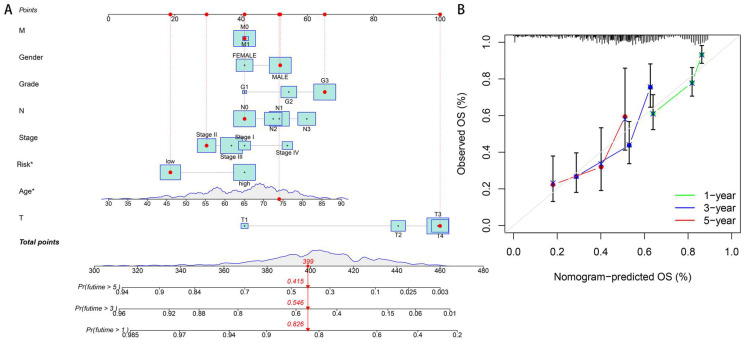
Nomogram development and validation. (**A**) An integrated prognostic nomogram incorporating clinicopathological parameters and disulfidptosis risk classification for the overall survival prediction of GA patients. In this figure, “Points” represent the score assigned to each variable, reflecting its weight contribution in the prediction model. “Total points” is the sum of all variable scores, used to predict survival probability. “M”, “N”, and “T” represent metastasis, lymph node, and tumor size status in TNM (tumor-node-metastasis) staging, respectively. “Pr (futime > x)” indicates the probability of survival beyond “x” years, where “futime” refers to follow-up time. The asterisks in “Risk*” and “Age*” indicate statistical significance in multivariate analysis (*p* < 0.05). Cyan boxes display the distribution density of variables, with box size reflecting patient concentration in that value range (central point = median). Red dots and connecting lines demonstrate the prediction pathway for an example patient, showing how to obtain scores from clinical features and derive the survival probability. Red arrows in the bottom subpanel indicate correspondence between the total number of points and 1-/3-/5-year survival probabilities. (**B**) A calibration plot assessment of nomogram accuracy. The top histogram displays the distribution of predicted probabilities, showing which intervals the predicted patient survival probabilities are concentrated in. The diagonal line represents the ideal prediction scenario; calibration curves approaching this line indicate higher predictive accuracy of the model.

**Figure 7 biomedicines-13-01224-f007:**
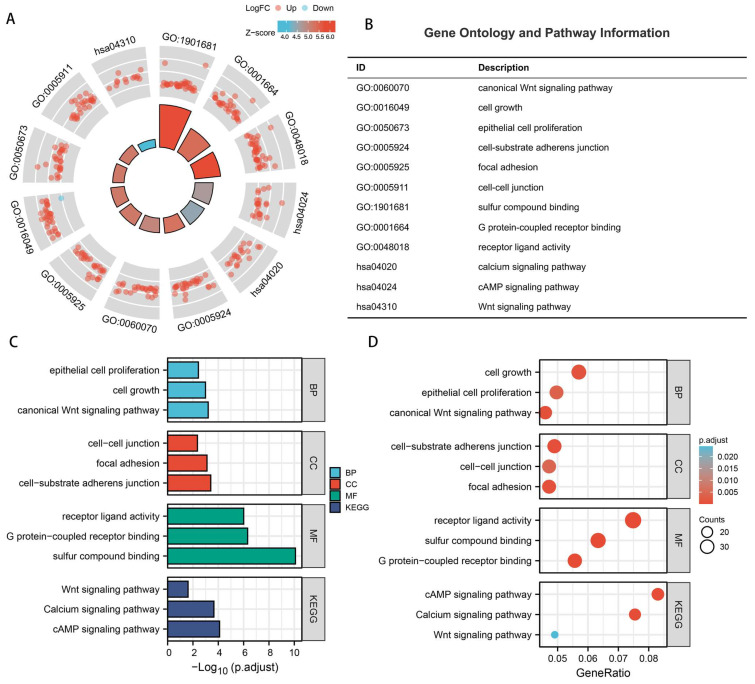
Functional pathway characterization between differential risk stratification cohorts. (**A**,**B**) GO enrichment analysis. The inner color-coded wedges represent different GO terms and KEGG pathways (such as GO:1901681, hsa04024, etc.). The size of each wedge reflects the number of enriched genes in the corresponding functional category—larger wedges indicate that more genes are enriched in that category. The colors of the wedges represent the Z-score values of the enrichment analysis, following the color scale shown in the upper right legend, ranging from 4.0 to 6.0 (light to dark), indicating the statistical significance level of enrichment. The red and blue spots displayed in the outer ring represent differentially expressed genes, with each spot representing one gene. As shown in the upper right legend, red spots indicate upregulated genes, while blue spots indicate downregulated genes. The position of these spots within each functional category sector indicates the biological processes or signaling pathways they participate in. (**C**,**D**) GO and KEGG enrichment analysis, including GO terms, biological process (BP), cellular component (CC), and molecular function (MF). ‘p.adjust’ refers to the *p*-value corrected using the Benjamini–Hochberg method, with lower values indicating the higher statistical significance of enrichment. ‘Counts’ represents the absolute number of genes from our gene set that are annotated to each specific pathway or functional category. ‘GeneRatio’ indicates the proportion of genes in our analyzed gene set that are associated with each pathway.

**Figure 8 biomedicines-13-01224-f008:**
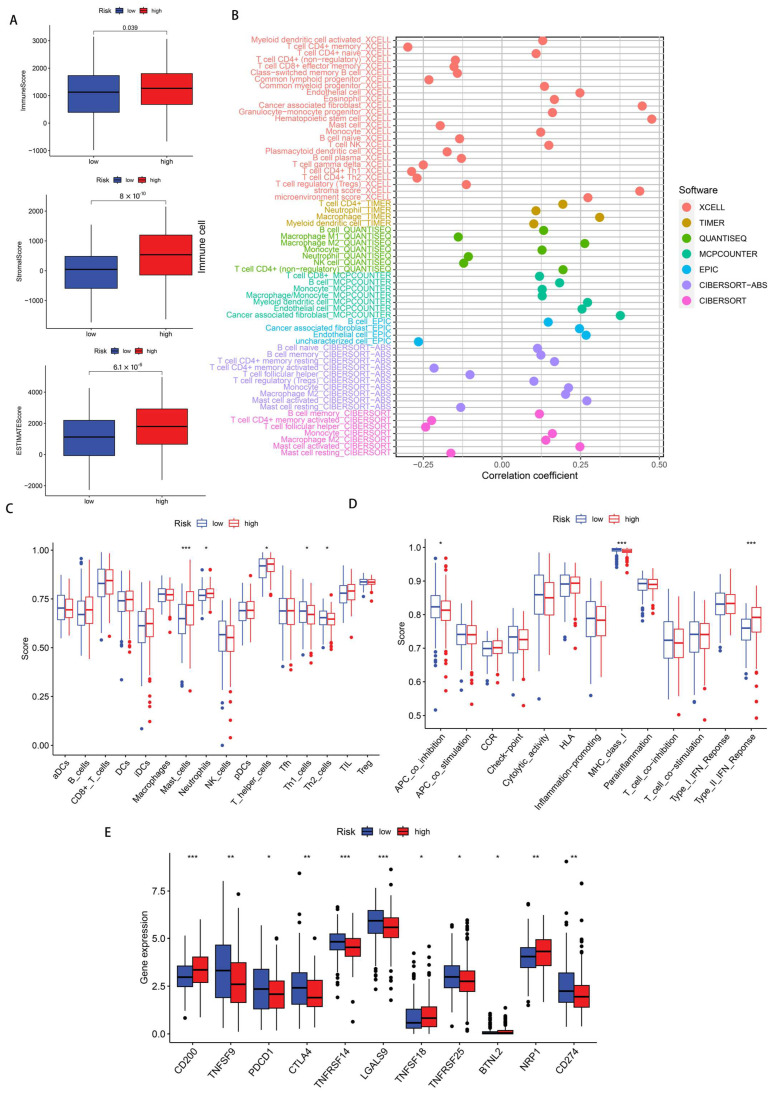
An evaluation of GA patients’ immunological microenvironments. (**A**) A comparative assessment of the immune infiltration metrics, stromal component quantification, and comprehensive microenvironment estimation of the elevated- and reduced-risk stratification cohorts. (**B**) A bubble plot visualization depicting the immunological cellular composition across risk-stratified populations, utilizing seven distinct computational algorithms. (**C**,**D**) A boxplot representation of the differential immune cell populations and immunological functional parameters of patient cohorts with contrasting risk classifications. (**E**) A boxplot of the expressional variation of ICPs between differential risk categories. * *p* < 0.05, ** *p* < 0.01, *** *p* < 0.001.

**Figure 9 biomedicines-13-01224-f009:**
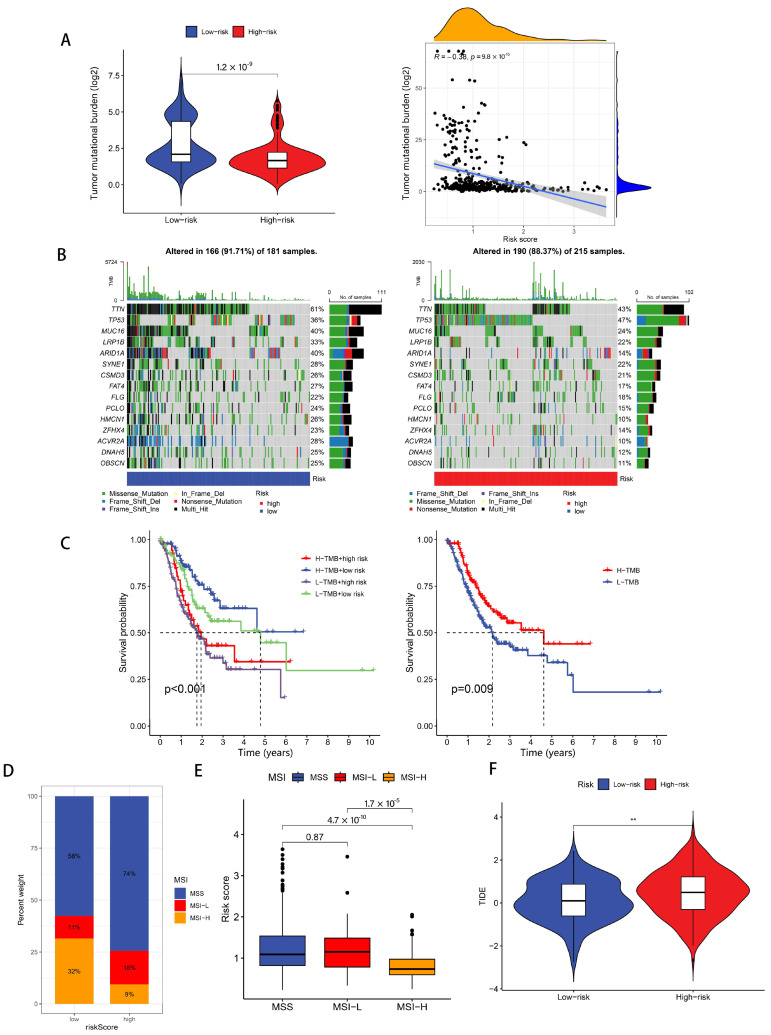
TMB, MSI, and TIDE analysis. (**A**,**B**) The differential somatic mutational landscape characterization of the elevated- and reduced-risk patient cohorts. Top yellow area in Figure A represents the density distribution curve of risk scores (x-axis). It shows the distribution of risk scores across all samples, indicating that most samples have risk scores concentrated in the lower range. Right-side blue area in Figure A represents the density distribution curve of tumor mutation burden (y-axis). It shows the distribution of TMB values across all samples, indicating that most samples have TMB values concentrated in the lower range, presenting a right-skewed distribution. (**C**) Kaplan–Meier survival analyses of the stratified GA patient subpopulations. (**D**,**E**) The association patterns between risk classification metrics and MSI status. (**F**) A comparative analysis of TIDE algorithmic scores between contrasting risk stratification categories. ** *p* < 0.01.

**Figure 10 biomedicines-13-01224-f010:**
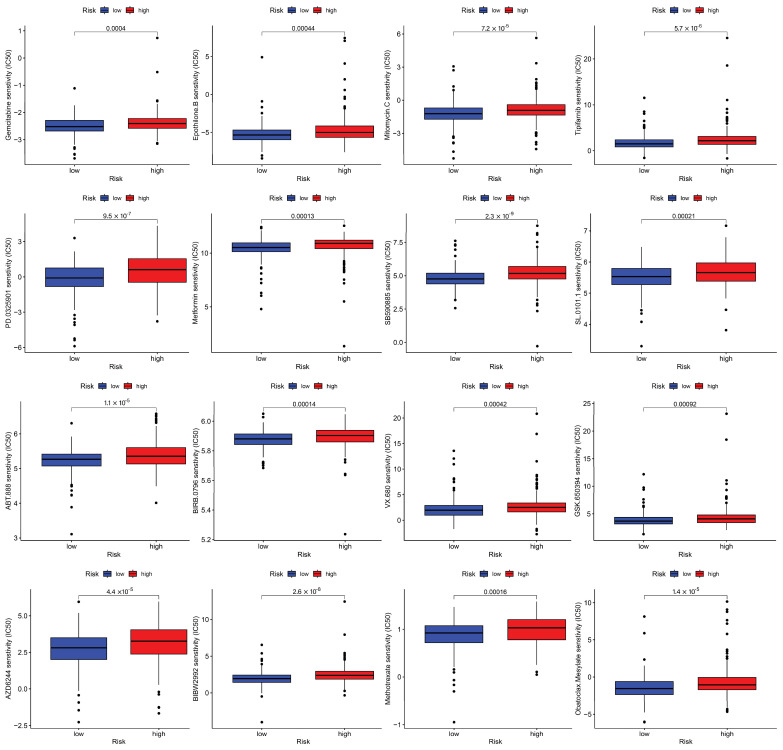
The sixteen therapeutic agents demonstrating enhanced efficacy in the low-disulfidptosis score group.

**Figure 11 biomedicines-13-01224-f011:**
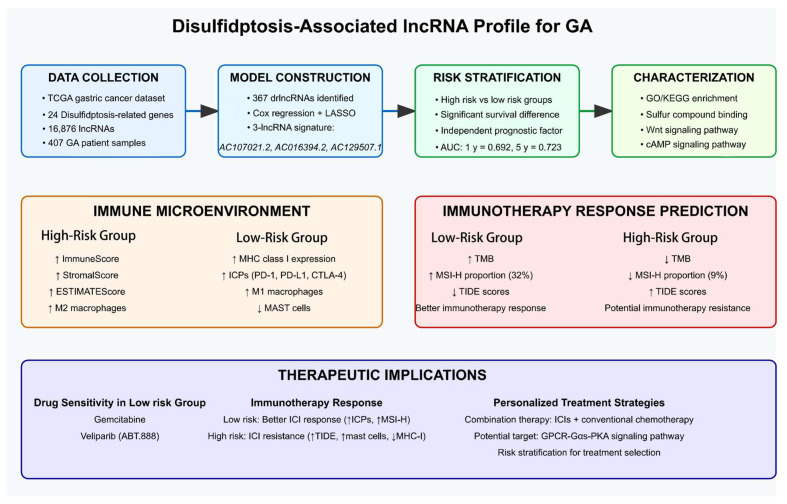
A comprehensive overview of the drlncRNA signature for prognosis prediction and personalized therapy in GA. The workflows of signature development, TME characterization, immunotherapy response prediction, and therapeutic implications are presented.

## Data Availability

The original contributions presented in the study are included in the article/Appendix A. Further inquiries can be directed toward the corresponding authors.

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
