# Peer review of "The Development and Assessment of a Unique Disulfidptosis-Associated lncRNA Profile for Immune Microenvironment Prediction and Personalized Therapy in Gastric Adenocarcinoma"

_biomedicines, 2025, doi:10.3390/biomedicines13051224_

Round 1
Reviewer 1 Report
Comments and Suggestions for Authors
I congratulate the team for the study. The results obtained seem to open up new opportunities for assessment and treatment. I hope you continue the study based on the premises highlighted in the article.
Author Response
Response to Reviewer Comments
Dear Editor and Reviewers,
We sincerely thank the reviewer for their thorough evaluation of our manuscript titled "Development and assessment of a unique disulfidptosis-associated lncRNA profile for immune microenvironment prediction and personalized therapy in gastric adenocarcinoma." We are pleased that the reviewer has found value in our research and has provided encouraging feedback. Below, we address each evaluation point:
Assessment Points
Reviewer Comment: Does the introduction provide sufficient background and include all relevant references? (Yes)
Author Response: We appreciate the reviewer's positive assessment of our introduction. We made a concerted effort to comprehensively review the current understanding of disulfidptosis as a novel cell death mechanism and the significance of long non-coding RNAs in gastric adenocarcinoma, ensuring inclusion of all relevant and recent literature.
Reviewer Comment: Is the research design appropriate? (Yes)
Author Response: We thank the reviewer for confirming the appropriateness of our research design. We carefully structured our methodological approach to ensure scientific rigor and reliability in developing and validating our disulfidptosis-associated lncRNA signature.
Reviewer Comment: Are the methods adequately described? (Yes)
Author Response: We are grateful for the reviewer's acknowledgment that our methods are adequately described. We endeavored to provide comprehensive methodological details to ensure reproducibility and clarity for readers.
Reviewer Comment: Are the results clearly presented? (Yes)
Author Response: We appreciate the reviewer's confirmation that our results are clearly presented. We have made significant efforts to present our findings in a structured and comprehensible manner, with appropriate figures and tables to facilitate understanding.
Reviewer Comment: Are the conclusions supported by the results? (Yes)
Author Response: We thank the reviewer for confirming that our conclusions are well-supported by our results. We have been careful to draw conclusions that are directly substantiated by our findings, avoiding overinterpretation while highlighting the potential clinical significance of our work.
Additional Reviewer Comment
Reviewer Comment: I congratulate the team for the study. The results obtained seem to open up new opportunities for assessment and treatment. I hope you continue the study based on the premises highlighted in the article.
Author Response: We sincerely thank the reviewer for their encouraging words. We fully agree that our findings on disulfidptosis-associated lncRNAs in gastric adenocarcinoma open up novel opportunities for patient assessment and potential therapeutic interventions. We are committed to continuing this line of research, building upon the foundation established in this study. Our future work will focus on validation in independent cohorts and deeper mechanistic investigations into how these lncRNAs influence gastric carcinoma progression and treatment response.
We once again express our gratitude to the reviewer for their positive evaluation and constructive feedback, which will help strengthen our manuscript and guide our future research directions.
Sincerely,
[Jiyue Zhu]
Reviewer 2 Report
Comments and Suggestions for Authors
Major points:
1) Please order the first three authors according to the alphabetical order with respect to their surnames and then first names.
2) The panels presented in Figures 2A–F, 3A–R, 4, 7A–E, 8A–F appear pixelated and are therefore difficult/impossible to read. Please replot using larger size and/or higher pixel density so that each individual text caption can be clearly comprehended. Feel free to split one multi-panel figure into two or several more figures.
3) The panels presented in Figures 6 and 9 are also somewhat pixelated.
4) Please provide all underpinning data sets as part of the supplementary material in an Excel spreadsheet format.
5) Please sketch either a graphical abstract or a new concluding figure (Figure 10) to neatly summarize the derived finding of this study.
Minor points:
1) Please change "progression" to "progression," (line 14).
2) Please replace "gastric cancer" with "gastric cancer (GC)" (line 14).
3) Please change "gastric cancer" to "GC" (lines 17, 19, 20, 53, 375, 472).
4) Please replace "long non-coding RNAs" with "lncRNAs" (lines 17, 77, 95, 114, 130, 222, 282, 372, 480).
5) Please define abbreviation for "ROC" (lines 22, 134), "GO" (line 24), "KEGG" (line 24), "MSI" (lines 26, 329), "TIDE" (line 26), "TME" (lines 32, 164), "ICPs" (lines 35, 326), "PDCD-1" (line 316), "PD-L1" (line 316), "CTLA-4" (line 316), "MSI-H" (line 343), "MSS" (line 346), "MSI-L" (line 346), "Gαs-PKA" (line 394), "PD-1" in "anti-PD-1" (line 417), "PDCD1" (line 427), "ICI" (line 431).
6) Please change "analysis" to "analysis," (line 22).
7) Please change "TMB" to "tumor mutational burden (TMB)" (line 26) and "tumor mutational burden (TMB)" to "TMB" (line 436).
8) Please replace "AC016394.2" with "AC016394.2," (line 27).
9) Please change "gastric carcinoma" to "GC" (lines 28, 42, 64, 79, 131, 221, 339, 481).
10) It is not exactly clear what the authors refer to as "sulfur compound" in "GO and KEGG pathway assessments revealed predominant enrichment within the elevated-risk cohort, particularly in pathways involving sulfur compound interactions, traditional WNT signaling mechanisms, cellular-substrate adhesion junctions, and cAMP signaling cascades, among others" (line 29), "The circle chart and corresponding notes (Figure 6A, B), column chart (Figure 6C) and bubble chart (Figure 6D) of GO and KEGG enrichment analysis revealed significant associations between these 3 drlncRNAs and sulfur compound binding, canonical WNT signaling pathway, cell-substrate adherens junction and cAMP signaling pathway" (line 292), "As anticipated, enrichment analysis revealed a strong association between disulfidptosis-related lncRNAs and sulfur compound binding" (line 388), "The identification of disulfidptosis revealed a promising therapeutic strategy for cancer treatment, wherein sulfur compound binding orchestrates this cell death process through the activation of specific signaling cascades[10] (line 389), and in Figure 6B–D? Please provide the definition for this term in the text.
11) Please replace "WNT" with "Wnt" (lines 31, 294).
12) Please change "cellular-substrate adhesion" to "cell-substrate adherens" (line 32).
13) Please replace "Tumor Immune Dysfunction and Exclusion" with "tumor immune dysfunction and exclusion" (line 37).
14) Please change "gastric carcinoma" to "gastric carcinoma (GC)" (line 47).
15) Please replace "annually[1]" with "annually [1]" (line 49).
16) Please change "gastric cancer-related" to "GC-related" (line 53).
17) Please replace "2040.[2]" with "2040 [2]" (line 53).
18) Please change "metastasis[3]" to "metastasis [3]" (line 55).
19) Please replace "years[4,5]" with "years [4,5]" (line 56).
20) Please change "malignancies[6]" to "malignancies [6]" (line 59).
21) Please replace "40-60%" with "40–60%" (line 60).
22) Please change "resistance[7]" to "resistance [7]" (line 61).
23) Please replace "STAD" with "GA" (lines 62, 82, 100, 103, 230, 486).
24) Please change "drlncRNAs" to "long non-coding RNAs associated with disulfidptosis (drlncRNAs)" (line 68).
25) Please change "DRGs (disulfidptosis-related genes)" to "disulfidptosis-related genes (DRGs)" (line 69).
26) Please replace "starvation[8]" with "starvation [8]" (line 73).
27) Please change "gastric adenocarcinoma" to "GA" (lines 81, 85, 92, 101, 109, 127, 173, 180, 188, 223, 289, 319, 352, 371, 373).
28) Please replace "patients.[12-15]" with "patients [12-15]" (line 81).
29) Please change "disulfidptosis-related lncRNA" to "drlncRNAs" (line 81).
30) Please replace "Figure-1" with "Figure 1" (line 91).
31) Please change "https://tcga-data.nci.nih.gov/tcga/" to "https://tcga-data.nci.nih.gov/tcga" (line 94).
32) Please replace "messenger RNAs" with "mRNAs" (line 95).
33) Please change "utilized" to "utilized the" (line 96).
34) Please replace "https://strawberryperl.com/" with "https://strawberryperl.com" (line 97).
35) Please provide reference for "24 DRGs were collected from the literature by Liu et al. mentioned before and Pearson correlation analysis was then performed to filter out drlncRNAs" (line 97).
36) Please change "The Cancer Genome Atlas" to "TCGA" (line 102).
37) Please replace "from" with "from the" (lines 103, 181).
38) Please change "https://www.tcia.at/" to "https://www.tcia.at" (line 104).
39) Please replace "Immune Dysfunction and Exclusion" with "immune dysfunction and exclusion" (line 104).
40) The black outline of the "STAD transcriptome profiling data in TCGA (n=448)" box seems to have two bottom regions with uneven shading in Figure 1. Please correct.
41) Please enlarge the diagram presented in Figure 1 so that it becomes easier to read.
42) Please replace "STAD" to "GA" (2x), "n=448" with "n = 448", "19938" with "19,938", "16876" with "16,876", "N=204" with "n = 204", "N=443" with "n = 443", "N=36" with "n = 36", "N=407" with "n = 407", "N=203" with "n = 203" in Figure 1.
43) Please define abbreviation for "OS", "LASSO" in "LASSO-penalized", "PCA", "GSEA" in the legend to Figure 1.
44) It is not exactly clear what the authors mean by "arlncRNAs" in "2.2. The development and verification of the arlncRNAs signature" (line 108)? Please fix in the text.
45) Please change "arlncRNAs" to "arlncRNA" (line 108).
46) Please replace "disulfidptosis-associated long non-coding RNAs" with "drlncRNAs" (line 113).
47) Please change "p<0.05" to "p < 0.05" (lines 115, 117, 211, 214, 216).
48) Please replace "7" with "seven" (line 117).
49) Please change "3" to "three" (lines 119, 245, 294, 473).
50) Please replace "disulfidptosis-related long non-coding RNAs" with "drlncRNAs" (line 119).
51) Please leave a gap between "the following mathematical expression:" (line 122) and "Risk score= lncRNA1exp × coef1 + lncRNA2exp × coef2 +… + lncRNAnexp × coefn." (line 123).
52) Please change "disulfidptosis-associated long non-coding RNA" to "drlncRNA" (lines 130, 281, 372).
53) Please replace ""survminer"" with ""survminer"," (line 133).
54) Please change "AUC (area under curve)" to "area under the curve (AUC)" (line 135).
55) Please replace "stage" with "stage," (line 143).
56) Please change "curves and AUC (area un-144 der curve)" to "curve and AUC" (line 144).
57) Please replace "values. (R packages "limma", "scatterplot3d", "survival", "survminer" and "timeROC" were used in this progress)" with "values (R packages "limma", "scatterplot3d", "survival", "survminer", and "timeROC" were used in this progress)." (line 145).
58) Please change "Calibration" to "calibration" (line 147).
59) Please replace "Gene Ontology" with "GO" (line 154) and "Gene Ontology" with "GO" (line 156).
60) Please change "infiltration" to "infiltration," (line 161).
61) Please replace "TME[16]" with "TME [16]" (line 164).
62) Please change "ESTIMATE" to "Estimation of STromal and Immune cells in MAlignant Tumor tissues using Expression data (ESTIMATE)" (line 164) and "Estimation of STromal and Immune cells in MAlignant Tumor tissues using Expression data (ESTIMATE)" to "ESTIMATE" (line 301).
63) Please replace "platforms:" with "platforms" (line 168).
64) Please change "MSI" to "MSI," (lines 179, 350).
65) Please replace "through" with "through the" (line 182).
66) Pleas change "15" to "fifteen" (lines 183, 333, 444).
67) Please replace "utilizing" with "utilizing the" (line 187).
68) Please change "Tumor Immune Dysfunction and Exclusion" to "tumor immune dysfunction and exclusion" (line 191).
69) Please replace "19938" with "19,938" (line 202).
70) Please change "16876" to "16,876" (line 202).
71) Please replace "disulfidptosis-related genes" with "DRGs" (line 203).
72) Please change "16876" to "16,876" (line 205).
73) Please replace "407 GC patients’ clinical information" with "clinical information of 407 GC patients" (line 208).
74) Please change "cox" to "Cox" (line 210).
75) Please replace "overall survival (OS)" with "OS" (line 213).
76) Please change "AC016394.2" to "AC016394.2," (line 215).
77) Please rewrite the "risk score formula" mentioned in "According to the previous calculation formula, risk score = AC107021.2exp × 0.66000986856416 + AC016394.2exp × -0.435804339506525 + AC129507.1exp × 0.636605050383588" using the equation function (line 217).
78) Please replace "disulfidptosis-related long non-coding RNA" with "drlncRNA" (line 222).
79) Please remove bold formatting from "," in "(C, D)" (lines 224, 298, 324).
80) Please replace "disulfidptosis-related gene" with "DRGs" (line 228).
81) Please change "as" to "as a" (line 232).
82) Please replace "group" with "groups" (line 233).
83) Please change "with" to "with the" (line 235).
84) Please replace "constructed" with something like "constructed by" (line 236).
85) Please change "disulfidptosis score" to "disulfidptosis score patients" (line 237).
86) Please replace "of" with "of the" (line 239).
87) Please change "p<0.001" to "p < 0.001" (line 239).
88) Please replace "high- risk" with "high-risk" (line 240).
89) Please change "were" to "was" (line 240).
90) Please replace "low- risk" with "low-risk" (line 240).
91) Please change "3E-H" to "3E–H" (line 243).
92) Please replace "3I-L" with "3I–L" (line 244).
93) Please change "in" to "in the" (lines 244, 275, 361).
94) Please replace "area under curve" with "AUC" (lines 248, 379).
95) Please change "Area under curve" to "AUC" (line 352).
96) Please replace "receiver operating characteristic" with "ROC" (line 252).
97) Please change "A-D" to "A–D" (line 259).
98) Please remove bold formatting from "-" in "(A-D)" (line 259).
99) Please replace "E-H, I-L" with "E–H, I–L" (line 261).
100) Please remove bold formatting from "-" (2x) and "," in "(E-H, I-L)" (line 261).
101) Please change "O-Q" to "O–Q" (line 265).
102) Please remove bold formatting from "-" in "(O–Q)" (line 265).
103) Please insert space between the legend to Figure 3 and the rest of the text.
104) Please indicate alphabetical designations for all panels depicted in Figure 4.
105) Please replace "risk***" with "Risk*", "Age***" with "Age*" "Pr( futime > 5)" with "Pr(futime > 5)", "Pr( futime > 3)" with "Pr(futime > 3)", "Pr( futime > 1)" with "Pr(futime > 1)" in the y-axis legend of Figure 5A.
106) "Pr( futime > 5)", "Pr( futime > 3)", "Pr( futime > 1)" descriptors seem to clash with their respective y-axes in Figure 5A. Please fix.
107) From the legend to Figure 5A is not explicitly clear what is the definition of "Points", "M", "N", "T", "Pr", and "futime"? It is also not clear what is the difference between "Points" and "Total points". Please correct in the text.
108) Furthermore, from the legend to Figure 5A is not evident why different cyan boxes are plotted with varying area sizes and what are the small black and large red circles as well as the black and dashed red connections?
109) Although "risk***" and "Age***" seem to be indicated by three asterisks in Figure 5A, the underlying meaning seems to be left unexplained.
110) Lastly, it is not clear what is the rationale for the three adjoined red arrows drawn between "Pr( futime > 5)", "Pr( futime > 3)", and "Pr( futime > 1)" in the "Total points" bottom subpanel of Figure 5A?
111) Similarly, the meaning of the top x-axis histogram and the gray diagonal line connection presented in Figure 5B seem to be elusive. Please explain in the respective figure legend.
112) The gray diagonal line connection presented in Figure 5B is barely visible. Please change its hue or color to a darker tone.
113) Please change "(Figure 6C)" to "(Figure 6C)," (line 292).
114) Please replace "junction" with "junction," (line 295).
115) Please change "Calcium" to "calcium" in Figure 6B.
116) Please remove bold formatting from "," in "(A, B)" (lines 298, 350).
117) Please replace "StromalScore(p=2.5e-10), ImmuneScore(p=0.039) and ESTIMATEScore(p=6.1e-06)" with "StromalScore (p = 2.5e-10), ImmuneScore (p = 0.039) and ESTIMATEScore (p = 6.1e-06)" (line 305).
118) Please change "between" to "between the" (line 307).
119) Please provide reference for each of the software mentioned in "The bubble plot visualization demonstrated that patient cohorts with elevated risk scores corresponded to enhanced immune cell infiltration density, such as CAF (Cancer associated fibroblast) in XCELL, MCPCOUNTER and EPIC, Macrophage 2 in QUANTISEQ, CIBERSORT and CIBERSORT-ABS" (line 307).
120) Please replace "The bubble" with "Bubble" (line 307).
121) Please change "Cancer associated" to "cancer-associated" (line 309).
122) Please replace "MCPCOUNTER" with "MCPCOUNTER," (line 310).
123) Please change "CIBERSORT" to "CIBERSORT," (line 310).
124) Please replace "MAST cells" with "mast cell" (line 313).
125) Please change "MHC class I" to "major histocompatibility complex class I (MHC-I)" (line 314) and "MHC class I (Major Histocompatibility Complex I)" to "MHC-I" (line 415).
126) Please change "(PD-L1)" to "(PD-L1)," (line 316).
127) Please replace "TIDE" with "and TIDE" (line 329).
128) Please change "and" to "and the" (lines 338, 343, 353).
129) Please replace "p=0.009" with "p = 0.009" (line 340).
130) Please change "“MSI-H”" to "MSI-H" (lines 343, 345).
131) Please replace "“MSS”" with "MSS" (line 346).
132) Please change "p=4.7e-10" to "p = 4.7e-10" (line 346).
133) Please replace "p=1.7e-05" with "p = 1.7e-05" (line 346).
134) Please change "Tumor Immune Dysfunction and Exclusion" to "TIDE" (line 346).
135) Please replace "among" with "among the" (line 348).
136) Please remove bold formatting from "," in "(D, E)" (line 353).
137) Please change "16" to "Sixteen" (line 361).
138) Please change "disulfidptosis-associated lncRNAs" to "drlncRNAs" (line 368).
139) Please replace "adenocarcinomas[17,18]" with "adenocarcinomas [17,18]" (line 370).
140) Please change "carcinoma[19,20]" to "carcinoma [19,20]" (line 371).
141) Please replace "our risk model's ability" with "the ability of our risk model" (line 377).
142) Please change "algorithm's" to "algorithm" (line 380).
143) Please replace "disulfidptosis-related lncRNAs" with "drlncRNAs" (line 388).
144) Please change "cascades[10]" to "cascades [10]" (line 392).
145) Please replace "GPCR (G protein coupled receptor)" with "G protein-coupled receptor (GPCR)" (line 393).
146) Please change "binding, cAMP" to "binding and the cAMP" (line 394).
147) Please format "+" in "CD8+" using superscript (line 395).
148) Please replace "immunotherapy[21]" with "immunotherapy [21]" (line 395).
149) Please change "(TAMs)[22-27]" to "(TAMs) [22-27]" (line 398).
150) Please replace "states:" with "states," (line 405).
151) Please change "M2[28]" to "M2 [28]" (line 405).
152) Please replace "M2" with something like "M2 macrophages" (lines 406, 409).
153) Please change "groups:" to "groups," (line 407).
154) Please replace "M1" with something like "M1 macrophage" (line 407).
155) Please change "M2" to something like "M2 macrophage" (line 408).
156) Please replace "M1 contributes significantly" with something like "M1 macrophages significantly contribute" (line 408).
157) Please change "cells[29,30]" to "cells [29,30]" (line 410).
158) Please replace "elucidates" with something like "substantiates" (line 411).
159) Please change "ssGSEA (single-sample Gene Set Enrichment Analysis)" to "ssGSEA" (line 414).
160) Please replace "blockade[31]" with "blockade [31]" (line 418).
161) Please change "patients[32]" to "patients [32]" (line 421).
162) Please replace "doesn't" with "does not" (line 422).
163) Please change "outcomes[33-36]" to "outcomes [33-36]" (line 426).
164) Please replace "for" with "for the" (line 431).
165) Please change "outcomes[37]" to "outcomes [37]" (line 432).
166) Please replace "immune checkpoint inhibitors" with "ICIs" (line 434).
167) Please change "accuracy[38-41]" to "accuracy [38-41]" (line 435).
168) Please replace "p=1.2e-09" with "p = 1.2e-09" (line 437).
169) Please change "found" to "found that" (line 440).
170) Please replace "cancers[42]" with "cancers [42]" (line 449).
171) Please change "protein[43]" to "protein [43]" (line 450).
172) Please replace "patients[44]" with "patients [44]" (line 452).
173) Please change "immune checkpoint inhibitors" to "ICI" (lines 455, 457).
174) Please replace "TMB[45-47]" with "TMB [45-47]" (line 456).
175) Please change "cancer[48]" to "cancer [48]" (line 467).
176) Please replace "ABT.888(veliparib)" with "ABT.888 (veliparib)" (line 468).
177) Please change "cancer[49]" to "cancer [49]" (line 470).
178) Please replace "framework's" with "framework" (line 478).
179) Please change "disulfidptosis-related lncRNA" to "drlncRNA" (line 483).
180) Please replace "X.Z." with "X.Z.," (lines 488, 489).
181) Please change "H.Z." to "H.Z.," (line 489).
Reviewer 3 Report
Comments and Suggestions for Authors
This manuscript by Zhu et al. aims to develop and assess a unique disulfidptosis-associated long noncoding RNA (lncRNA) profile for immune microenvironment prediction and personalized therapy in gastric adenocarcinoma using publicly available data sets acquired from TCGA. To this end, the authors start with 448 publicly available gastric adenocarcinoma transcriptomic data sets and conduct their analyses with the data that belong to 407 STAD patients. Using different regression analysis methods, the authors then develop a disulfidptosis-related lncRNA (drlncRNA) signature. Interestingly, a model composed of 3 drlncRNAs was successfully established. Gene Ontology (GO) and KEGG assessments were used to uncover the molecular functions associated. Finally, based on the drlncRNA signature, the authors were able to examine the efficacy of some therapeutic agents, such as gemcitabine and veliparib.
Despite the use of immunotherapy to treat gastric cancer, therapeutic resistance requires the discovery of novel drug targets. Disulfidptosis as a caspase-independent cell death mechanism holds promise in this respect. The authors provide an interesting model to assess the immune microenvironment for personalized therapy. I believe the findings from this work could be helpful to researchers working in this field. I recommend the following points for consideration to improve the manuscript.
Major points:
- Lines 123-125: Please present the formula in a mathematical formula. The same applies to lines 217-218.
- Lines 201-202: Please clarify what “expression data” means. Read counts?
- Line 204: Why did the authors choose a low threshold level of 0.4? I believe it is a bit low.
- The resolution of most figures (esp 2 through 9) must be improved. It is difficult to read the legends. Also, the captions should be explained.
- Section 3.4: The authors mention gene set enrichment analyses (GSEA) both in the method and discussion sections (e.g. lines 413-414), but I do not see the results being presented in the Results section.
- Section 3.7 and Figure 9: The findings from this section should be detailed. How many drugs were screened? How many were effective (16??)? Why were two drugs prioritized?
Minor points:
- There are several typos. (a) Line 90: “date” should be “data”; (b) line 147: “nomagram” should be “nomogram”
- Please abbreviate a phrase once and use the abbreviation throughout the manuscript. E.g., tumor mutational burden (TMB) in lines 102, 181, 339 and 436.
Round 2
Reviewer 2 Report
Comments and Suggestions for Authors
Major points:
1) With regards to the first authorship, the authors are advised to choose from the two following options:
a) either arrange all first authors according to the alphabetical order (surname, first name)
b) or delegate only one first author
2) Figures 2B–D; 3A–R; 4A–F; 5A; 7A–D; 8A–F; 9 are still way too pixelated. It is imperative that each single text element within a figure can be clearly read by the reader without having to resort to extreme focus.
Please combine the following approaches:
a) increase pixel density/resolution so that every caption whether it is a gene name, graph axis title, or just an asterisk can be sharply distinguished
b) increase panel size to allow for in-depth inspection also for non-digital format (hardcopy printout) readers
c) divide the copious panels among several figures, which can still fall within the main text or, depending on priority, within the supplementary material
3) Figures 2A,E,F; 6A; 7E appear somewhat pixelated.
4) Please provide the supplementary data material as part of the final manuscript package.
Minor points:
1) Please replace "GC" with "gastric carcinoma (GC)" (line 14).
2) Please change "ROC (Receiver Operating Characteristic)" to "receiver operating characteristic (ROC)" (lines 21, 132).
3) Please replace "GO (Gene Ontology) and KEGG (Kyoto Encyclopedia of Genes and Genomes)" with "Gene Ontology (GO) and Kyoto Encyclopedia of Genes and Genomes (KEGG)" (line 24).
4) Please change "MSI (microsatellite instability), and TIDE (Tumor Immune Dysfunction and Exclusion)" to "Tumor immune dysfunction and exclusion (TIDE), and microsatellite instability (MSI)" (line 27).
5) Please replace "3" with "three" (line 29).
6) Please change "TME (tumor microenvironment)" to "Tumor microenvironment (TME)" (line 34).
7) Please replace "ICPs (immune checkpoints)" with "immune checkpoints (ICPs)" (line 37).
8) Please change "microsatellite instability-high" to "MSI-high" (line 39).
9) Please replace "blockade agents (ICBs)" with "blockade (ICB) agents" (line 59).
10) Please replace "immune checkpoint-targeted" with "immune checkpoint (ICP)-targeted" (line 61).
11) Please provide reference for "Initial characterization by Liu and colleagues established disulfidptosis as a distinct cellular death mechanism triggered by disulfide bond overaccumulation within cells expressing elevated levels of SLC7A11" (line 67).
12) Please replace "drlncRNAs[8]" with "drlncRNAs [8]" (line 99).
13) Please remove blue color and underline formatting from "http://tide.dfci.harvard.edu" (line 106).
14) The "Exclusion" label gives rather a pixelated impression in Figure 1. Please fix.
15) Please change "used" to "used the" (line 131).
16) Please replace "utilizing" with "utilizing the" (line 148).
17) Please change "GO" to "Gene Ontology (GO)" (line 153).
18) Please replace "TME (tumor microenvironment)" with "tumor microenvironment (TME)" (line 162).
19) Please change "through" to "through the" (lines 164, 173, 175).
20) Please replace "tumor microenvironment" with "TME" (lines 165, 325, 423).
21) Please change "immune checkpoint" with "ICP" (line 175).
22) Please replace "immune checkpoints" with "ICPs" (line 176).
23) Please change "group" to "groups" (line 177).
24) "We first implemented univariate Cox regression analysis and preliminary screening out 29 drlncRNAs" (line 208) does not seem to be correct with respect to "preliminary screening out 29 drlncRNAs". Please rephrase.
25) Please convert "Risk score = 0.660 × 215
AC107021.2exp − 0.436 × AC016394.2exp + 0.637 × AC129507.1exp" to a standard formula using the Equation function (line 215).
26) Please replace "AC129507.1exp" with "AC129507.1exp." (line 216).
27) Please change "AC129507.1" to "AC129507.1 lncRNAs" (line 232).
28) Please replace "higher disulfidptosis" with something like "high-disulfidptosis" (line 232).
29) Please change "low disulfidptosis" to "low-disulfidptosis" (line 233).
30) Please replace "of" with "of the" (lines 234, 360).
31) Please change "drlncRNAs signature" to "drlncRNA signatures" (line 241).
32) Please remove bold formatting from "," in "E–H, I–L" (line 256).
33) Please replace "in" with "in the" (lines 257, 329).
34) Please change "sufficient" to something like "sufficient number of samples" (line 272).
35) Please move "In this figure, 'Points' represents the score assigned to each variable, reflecting its weight contribution in the prediction model; 'Total points' is the sum of all variable scores, used to predict survival probability; 'M', 'N', 'T' represent metastasis, lymph node, and tumor size status in TNM staging, respectively; 'Pr (futime > x)' indicates the probability of survival beyond x years; 'futime' refers to follow-up time. The asterisks in 'Risk*' and 'Age*' indicate statistical significance of these variables in multivariate analysis (p < 0.05). Cyan boxes display the distribution density of variables in the study population, with box size reflecting patient concentration in that value range, and the central point representing the median of the variable. Red dots and their connecting lines demonstrate the prediction pathway for an example patient, showing how to obtain corresponding scores from clinical features, accumulate them to get a total score, and derive survival probability. Red arrows in the bottom subpanel indicate the correspondence between total points and 1-year, 3-year, and 5-year survival probabilities." to the legend of Figure 5A.
36) Please define abbreviation for "TNM" (line 282).
37) Please replace "derive" with "derive the" (line 290).
38) Please change "the correspondence between total" to something like "correspondence between the total number of" (line 291).
39) Please move "The top histogram displays the
distribution of predicted probabilities, showing which intervals the patients' predicted
survival probabilities are concentrated in. The diagonal line represents the ideal predic- 295
tion scenario; calibration curves approaching this diagonal line indicate higher predictive
accuracy of the model." (line 293) to the legend of Figure 5B.
40) Please replace "and" with "and the" (line 306).
41) From the legend to Figure 6A is not evident what is the exact meaning of the depicted circle chart? Namely, what does the size of the inner color-coded wedges indicate? What is the plotted quantity for the red spots displayed in the outer ring and what is its scale?
42) Please define abbreviations for "BP", "CC", and "MF" in the legend to Figure 6C,D.
43) From the legend to Figure 6D is not clear the definition of "p. adjust", "Counts", and "GeneRatio". Please fix.
44) Please change "elevated risk" to "elevated-risk" (line 328).
45) Please replace "reduced risk" with "reduced-risk" (lines 328, 361).
46) Please change "(p = 0.039)" to "(p = 0.039)," (line 329).
47) Please replace "CAF (cancer-associated fibroblast)" with "cancer-associated fibroblast (CAF )" (line 333).
48) Please change "MCPCOUNTER, [22]" to "MCPCOUNTER [22]," (line 334).
49) Please replace "[23], Macrophage 2" with something like "[23], and macrophage M2" (line 334).
50) Please change "Immune checkpoint" to "ICP" (line 339).
51) Please change "PDCD-1 (programmed cell death protein 1)" to "programmed cell death protein 1 (PDCD-1)" (line 340).
52) Please replace "(PD-L1) (programmed cell death ligand 1)" with "(programmed cell death ligand 1) (PD-L1)" (line 341).
53) Please change "CTLA-4 (cytotoxic T lymphocyte-associated antigen 4)" to "cytotoxic T lymphocyte-associated antigen 4 (CTLA-4)" (line 341).
54) Please replace "ICPs (immune checkpoint)" with "ICPs" (line 350).
55) Please change "MSI (microsatellite instability)" to "microsatellite instability (MSI)" (line 353).
56) Please replace "high disulfidptosis" with "high-disulfidptosis" (lines 359, 431).
57) Please change "high risk" to "high-risk" (line 360).
58) Please replace "macrophage" with "macrophages" (line 429).
59) Please change "anti-PD-1 (programmed cell death protein 1)" to "anti-programmed cell death protein 1 (PD-1)" (line 436).
60) Please replace "ICP blockade" with "ICB" (line 437).
61) Please replace "PDCD1 (programmed cell death protein 1)" with "programmed cell death protein 1 (PDCD1)" (line 446).
62) Please change "ICI (immune checkpoint inhibitor)" to "immune checkpoint inhibitor (ICI)" (line 451).
63) Please replace "characterize" with "characterize the" (line 461).
64) Please change "immune checkpoint blockade" to "ICB" (line 480).
65) Please change "classification, while" to "classification while" (line 482).
66) Please replace "Gastric Cancer Dataset" with "gastric cancer dataset", "16876" with "16,876", "STAD" with "GA", "vs Low risk" with "vs low risk", "1y=0.692, 5y=0.723" with "1 y = 0.692, 5 y = 0.723", "Low Risk" with "Low risk", "High Risk" with "High risk" in Figure 10.
Author Response
Dear Reviewer 2,
Thank you very much for your detailed review and valuable suggestions regarding our manuscript. We have made the following changes as per your recommendations:
- Regarding the first authorship issue, we have chosen to designate only one first author (Jiyue Zhu).
- To address the pixelation issues in the figures, we have implemented several improvements:
- Increased the resolution of all images
- Enlarged the panel sizes of the figures
- Moved the univariate forest plot from original Figure 2 and original Figure 4 to the supplementary materials
- Reorganized the original Figure 3 into three separate figures (now Figures 3, 4, and 5)
- Regarding the supplementary data materials, when submitting the revised manuscript, we discovered that files submitted to reviewers can only be in Word or PDF format. Our supplementary tables S1-S9 are Excel files and quite large, making it impossible to upload them directly to you. Therefore, we have included all supplementary materials in the link submitted to the editor, compressed in RAR format. We apologize for any inconvenience this may cause.
- Regarding the 66 minor points you suggested, we have addressed all of them according to your requirements, including:
- Corrected all abbreviation expressions, such as changing "GC" to "gastric carcinoma (GC)"
- Fixed the pixelation issue with the "Exclusion" label in Figure 1
- Adjusted text formatting, including removing the blue color and underline formatting from URLs
- Reformatted equations and statistical symbols
- Moved the detailed descriptions for Figures 5A and 5B to their respective legends
- Defined abbreviations such as "BP," "CC," and "MF" in Figures 6C and 6D
- Corrected the definitions of "p. adjust," "Counts," and "GeneRatio" in Figure 6D
- Corrected the expressions in Figure 10
We believe these modifications have effectively addressed the issues you raised. We appreciate your guidance, which has significantly improved the quality of our manuscript. If you have any further suggestions, please do not hesitate to let us know.
Sincerely,
[Jiyue Zhu]
Reviewer 3 Report
Comments and Suggestions for Authors
The authors have addressed all the concerns raised in the first round of review. However, I would like the authors to incorporate their answers into the manuscript, especially for comments #2 and 6.
Comment 2: Lines 201-202: Please clarify what “expression data” means. Read counts?
Comment 6: Section 3.7 and Figure 9: The findings from this section should be detailed. How many drugs were screened? How many were effective (16??)? Why were two drugs prioritized?
I am satisfied with the answers, but it would be better to entegrate them into the manuscript to eliminate ambiguity.
Author Response
Dear Reviewer 3,
Thank you very much for your thorough review and valuable suggestions. We have integrated our answers into the revised manuscript as you suggested, particularly addressing your comments #2 and #6:
- Regarding comment #2 (Lines 201-202: clarification of "expression data"): We have added the following explanation in the methods section: "Using Strawberry Perl software, we processed the TCGA transcriptome data (raw gene count data generated using the STAR alignment tool, with file name suffix: 'augmented_star_gene_counts.tsv') and gained the expression data of 19,938 mRNAs and 16,876 lncRNAs. This dataset contains unnormalized read counts mapped to each gene, representing the original number of reads aligned to each gene region before any normalization based on gene length or sequencing depth (such as FPKM or TPM)."
- Regarding comment #6 (Section 3.7 and Figure 9: detailed findings): We have supplemented Section 3.7 with additional details on the number of effective drugs in the low-risk group (17 drugs) and clarified the specific rationale for prioritizing these two drugs, thereby eliminating any potential ambiguity.
We greatly appreciate your constructive feedback, which has made our manuscript more clear and comprehensive. If you have any further questions, please do not hesitate to let us know.
Sincerely,
[Jiyue Zhu]
Round 3
Reviewer 2 Report
Comments and Suggestions for Authors
Major points:
1) The numbers just on the right side of the y-axis title of Figure 2C are very difficult to read as they are pixelated. Please replot this panel using higher pixel density so that all individual text/captions can be accessed by the readers.
2) Figure 2C appears slightly pixelated.
3) Please remove the old overlapping Figures 3, 7–10.
4) Please either remove the figure panel found between lines 298 and 299 or turn it back into a full-fledged figure.
Minor points:
1) Please change "upregulate following the" to something like "are upregulated by" (line 76).
2) Please replace "in" with "of" (line 83).
3) Please change "from the literature by Liu et al. mentioned before" to either "from the study by Liu et al." or "from Liu et al." (line 100).
4) Please replace "remained eventually" with "remained" (line 102).
5) Please change "from" to "from the" (line 104).
6) Please replace "OS (Overall Survival)" with "overall survival (OS)", "(LASSO) Least absolute shrinkage and selection operator -penalized" with "Least absolute shrinkage and selection operator (LASSO)-penalized" in Figure 1.
7) Please define abbreviation for "TCIA" (line 106), "FPKM" (line 206), "TPM" (line 206).
8) Please move "Figure 1. The methodological framework of the present study." (line 109) to line 105.
9) Please change "train" to "training" (lines 114, 135, 239, 244).
10) Please replace "least" with "Least" (line 117).
11) Please provide gap between "Risk Score = Σ (i=1 to n) Coef(i) × Expr(i)" (line 125) and line 126.
12) Please change "i=1" to "i = 1" (line 125).
13) Please provide gap between "Expr(i) is the expression level of the i-th lncRNA" (line 127) and line 128.
14) Please replace "test set, entire set and" with something like "the test set, and the entire set as well as" (line 135).
15) Please change "characteristics" to "characteristics," (line 136).
16) Please replace "value" with "values" (line 136).
17) Please change "R" to "the R" (line 145).
18) Please replace "1-, 3-, and 5-year" with "1, 3, and 5 year" (line 150).
19) Please change "one-year, three-year, and five-year" to "1, 3, and 5 year" (lines 151, 305, 475, 480).
20) Please replace "TME" with "the tumor microenvironment" (line 161).
21) Please change "ESTIMATE" to "The Estimation of STromal and Immune cells in MAlignant Tumor tissues using Expression data (ESTIMATE)" (line 162) and "Estimation of STromal and Immune cells in MAlignant Tumor tissues using Expression data (ESTIMATE)" (line 164).
22) Please replace "of" with "of the" (line 164).
23) Please change "platforms" to "platforms," (line 170).
24) Please replace "utilizing" with "utilizing the" (line 175).
25) Please change "using" to "using the" (line 177).
26) Please replace "ICP" with something like "an" (line 178).
27) Please change "microsatellite instability" to "the microsatellite instability (MSI)" (line 190) and "microsatellite instability (MSI)" to "MSI" (line 421).
28) Please replace "|Pearson R|>0.4" with "|Pearson R| > 0.4" (line 208).
29) Please change "and corresponding" to "and the corresponding" (line 219).
30) From "Risk Score = 0.660 × AC107021.2_exp 0.436 × AC016394.2_exp + 0.637 × AC129507.1_exp" is not clear what is the meaning of "AC107021.2_exp", "AC016394.2_exp", "AC129507.1_exp"? Please correct the formula or explain in the text.
31) Please remove italics formatting from "Risk Score" (line 223).
32) Please replace "Parital Likelihood Deviance" with "Partial likelihood deviance" in the y-axis title of Figure 2B.
33) Please either change "Log(λ)" to "Log Lambda" in Figure 2B or "Log Lambda" to "Log (λ)" in Figure 2C.
34) Please replace "1.935(1.053–3.556)" with "1.935 (1.053–3.556)", "0.647(0.467–0.896)" with "0.647 (0.467–0.896)", "1.890(1.098–3.254)" with "1.890 (1.098–3.254)" in Figure 2D.
35) From the legend to Figure 2B is not clear the quantity of the top x axis and the meaning of the two horizontal dashed lines?
36) From the legend to Figure 2B,C is not clear the definition of the parameter lambda?
37) Please define abbreviation for "CI" in the legend to Figures 2D and S1.
38) Please change "(A–C):" to "(A–C)" (line 264).
39) Please remove bold formatting from "(" and ")" in "(A–C)" (line 264).
40) Please remove bold formatting from "(" and ")" in "(D)" (line 265).
41) Please remove bold formatting from "(" and ")" in "(E)" (line 266).
42) Please replace "1-year" with "1 year" (lines 270, 271, 272, 285).
43) Please change "3-year" to "3 year" (lines 270, 271, 272, 285).
44) Please replace "5-year" with "5 year" (lines 270, 271, 273, 285).
45) Please remove "The top histogram displays the distribution of predicted probabilities, showing which intervals the patients' predicted survival probabilities are concentrated in. The diagonal line represents the ideal prediction scenario; calibration curves approaching this diagonal line indicate higher predictive accuracy of the model." (line 320).
46) Please change "patients' predicted" to "predicted patient" (lines 321, 347).
47) Please remove bold formatting from "(" and ")" in "(A)" (line 325).
48) Please replace "figure:" with "figure," (line 328).
49) Please change "represents" to "represent" (line 329).
50) Please replace "Tumor-Node-Metastasis" with "tumor-node-metastasis" (line 332).
51) Please remove bold formatting from "(" and ")" in "(B)" (line 345).
52) Please change "histogram: Displays" to "histogram displays" (line 346).
53) Please replace "line: Represents" with "line represents" (line 348).
54) From "The circle chart and corresponding notes (Figure 7A, B), column chart (Figure 7C), and bubble chart (Figure 7D) of GO and KEGG enrichment analysis revealed significant associations between these three drlncRNAs and the sulfur compound binding, canonical Wnt signaling pathway, cell-substrate adherens junction, and cAMP signaling pathway" (line 351) is not explicitly clear what the authors mean by "these three drlncRNAs"?
55) Please provide reference for "At the molecular level, "sulfur compound binding" primarily manifests as proteins interacting with sulfur-containing molecules through disulfide cross-linking (such as SLC7A11 binding to cystine), enzyme-substrate binding (such as thioredoxin reductase TXNRD1 binding to oxidized TXN), and antioxidant regulation (such as covalent 364 binding of glutathione transferase GST to GSH)" (line 361).
56) Please remove bold formatting from "(" and ")" in "(A, B)" (line 373).
57) Please change "up-regulated" to "upregulated" (line 381).
58) Please replace "down-regulated" with "downregulated" (line 381).
59) Please remove bold formatting from "(" and ")" in "(C, D)" (line 383).
60) Please change "terms: BP (Biological Process), CC (Cellular Component), MF (Molecular Function)" to "terms, biological process (BP), cellular component (CC), molecular function (MF)" (line 384).
61) Please replace "difference of" with "difference of the" (line 391).
62) Please change "in" to "in the" (line 397).
63) Please replace "with" with "with the" (line 433).
64) Please change "Tumor tmbation burden" to "Tumor mutational burden" in the y-axis title of Figure 9A.
65) Please move "Figure 10. Sixteen therapeutic agents demonstrating enhanced efficacy in the low 457 disulfidptosis scores group." (line 457) to line 452.
66) Please replace "drlncRNAs" with "drlncRNA" (line 466).
67) Please change "investigated" to something like "investigated the role of" (line 468).
68) Please replace "status, illustrated" with "status illustrated" (line 478).
69) Please align all entries horizontally so that all bullet points "•" become exactly positioned on top of each other in the "DATA COLLECTION", "MODEL CONSTRUCTION", "RISK STRATIFICATION", and "CHARACTERIZATION" lists in Figure 11.
70) Please move "Figure 11. Comprehensive overview of the drlncRNA signature for prognosis prediction and personalized therapy in GA. The workflow of signature development, TME characterization, immunotherapy response prediction, and therapeutic implications are presented." (line 592) to line 590.
71) Please change "Hazard Ratio(95 %CI)" to "Hazard Ratio (95 %CI)" in Figure S1.
72) Please replace "Hazard Ratio(95 %CI)" to "Hazard Ratio (95 %CI)" also in the x-axis title of Figure S1.
73) Please change "age>65" to "age > 65", "age<=65" to "age ≤ 65" in Figure S2A.
74) Please replace "Time(years)" with "Time (years)" in the x-axis title of Figure S2A–F.
Author Response
Response to Reviewer's Comments
Main Issues Addressed
- Figure 2C Y-axis right side number readability and pixelation: We have redrawn this figure by reducing the font size and increasing the overall size of Figure 2C, ensuring all text is clearly legible and eliminating pixelation issues. The original Figure 2C has been moved to Supplementary Figure 3.
- Removal of overlapping images: We have deleted the old overlapping Figures 3, 7-10 to avoid confusion, and apologize for the overlapping issues in the previous submission.
- Figure plate between lines 298-299: This was an image duplication error. The original image has been placed in Supplementary Figure 2.
Minor Issues Addressed (Summary) We have made the following modifications to address minor issues, categorized for conciseness:
- Text Revisions:
- Modified wording to improve clarity and grammatical accuracy, e.g., changing "upregulate following the" to "are upregulated by", "in" to "of", "remained eventually" to "remained".
- Standardized terminology format, e.g., changing "1-, 3-, and 5-year" to "1, 3, and 5 year", "TME" to "the tumor microenvironment", "Tumor-Node-Metastasis" to "tumor-node-metastasis".
- Corrected capitalization and formatting, e.g., "least" to "Least", "Parital Likelihood Deviance" to "Partial likelihood deviance" (Figure 2B Y-axis title), and removed italics from "Risk Score".
- Abbreviation Definitions and Terminology Clarification:
- Defined abbreviations: TCIA, FPKM, TPM, and CI (in Figure 2D and S1 captions).
- Expanded terms, e.g., spelling out "ESTIMATE" in full, changing "microsatellite instability" to "the microsatellite instability (MSI)".
- Figure and Formula Clarification:
- Explained the meaning of "AC107021.2_exp", "AC016394.2_exp", and "AC129507.1_exp" in the Risk Score formula.
- Standardized figure symbols, e.g., changing "Log(λ)" to "Log Lambda" in Figure 2B, correcting confidence interval format in Figure 2D (e.g., "1.935(1.053–3.556)" to "1.935 (1.053–3.556)").
- Added explanations for the X-axis values, meaning of dashed lines, and definition of parameter lambda in Figure 2B and 2C legends.
- Format and Structure Adjustments:
- Adjusted spacing and punctuation, e.g., adding blank lines after formulas, correcting "(A–C):" to "(A–C)", removing bold formatting from parentheses.
- Repositioned figure captions to improve clarity, e.g., adjusting caption positions for Figures 1, 10, and 11.
- Aligned bullet points in Figure 11 for "DATA COLLECTION", "MODEL CONSTRUCTION", "RISK STRATIFICATION", and "CHARACTERIZATION" lists.
- Specific Clarifications:
- Added references for the description of sulfur compound binding.
- Clarified the identity of "these three drlncRNAs" in Figure 7.
- Corrected axis titles, e.g., "Tumor tmbation burden" to "Tumor mutational burden" (Figure 9A Y-axis), "Time(years)" to "Time (years)" (Figure S2A–F X-axis).
- Other Corrections:
- Removed redundant text and optimized sentence structure, e.g., changing "patients' predicted" to "predicted patient".
- Standardized mathematical symbols, e.g., "|Pearson R|>0.4" to "|Pearson R| > 0.4", "i=1" to "i = 1".
We believe these modifications comprehensively address all comments and improve the quality of our manuscript. Thank you again for your feedback, and we apologize for the image overlapping issues.
Sincerely,
[Jiyue Zhu]